# Genome-wide association study identifies the *SERPINB* gene cluster as a susceptibility locus for food allergy

Ingo Marenholz[1,2], Sarah Grosche[1,2], Birgit Kalb[1,2,3], Franz Rüschendorf [1], Katharina Blümchen[4], Rupert Schlags[5], Neda Harandi[5], Mareike Price[6], Gesine Hansen[6], Jürgen Seidenberg[7], Holger Röblitz[8], Songül Yürek[3], Sebastian Tschirner[3], Xiumei Hong[9], Xiaobin Wang[9], Georg Homuth[10], Carsten O. Schmidt[11], Markus M. Nöthen[12], Norbert Hübner[1], Bodo Niggemann[3], Kirsten Beyer[3] & Young-Ae Lee[1,2]

Genetic factors and mechanisms underlying food allergy are largely unknown. Due to heterogeneity of symptoms a reliable diagnosis is often difficult to make. Here, we report a genome-wide association study on food allergy diagnosed by oral food challenge in 497 cases and 2387 controls. We identify five loci at genome-wide significance, the clade B serpin (SERPINB) gene cluster at 18q21.3, the cytokine gene cluster at 5q31.1, the filaggrin gene, the *C11orf30/LRRC32* locus, and the human leukocyte antigen (HLA) region. Stratifying the results for the causative food demonstrates that association of the HLA locus is peanut allergy-specific whereas the other four loci increase the risk for any food allergy. Variants in the SERPINB gene cluster are associated with *SERPINB10* expression in leukocytes. Moreover, SERPINB genes are highly expressed in the esophagus. All identified loci are involved in immunological regulation or epithelial barrier function, emphasizing the role of both mechanisms in food allergy.

[1] Max-Delbrück-Center (MDC) for Molecular Medicine, 13125 Berlin, Germany. [2] Clinic for Pediatric Allergy, Experimental and Clinical Research Center, Charité University Medical Center, 13125 Berlin, Germany. [3] Department of Pediatric Pneumology and Immunology, Charité University Medical Center, 13353 Berlin, Germany. [4] Department of Allergy, Pulmonology and Cystic Fibrosis, Children's Hospital, Goethe University, 60590 Frankfurt am Main, Germany. [5] Department of Pediatric Pneumology and Allergology, Wangen Hospital, 88239 Wangen, Germany. [6] Department of Pediatric Pneumology, Allergology and Neonatology, Hannover Medical School, 30625 Hannover, Germany. [7] Department of Pediatric Pneumology and Allergology, Neonatology and Intensive Care, Medical Campus of University Oldenburg, 26133 Oldenburg, Germany. [8] Department of Pediatrics and Adolescent Medicine, Sana Klinikum Lichtenberg, 10365 Berlin, Germany. [9] Department of Population, Family and Reproductive Health, Center on the Early Life Origins of Disease, Johns Hopkins University Bloomberg School of Public Health, Baltimore, MD 21205, USA. [10] Department of Functional Genomics, Interfaculty Institute for Genetics and Functional Genomics, University Medicine and Ernst-Moritz-Arndt-University Greifswald, 17487 Greifswald, Germany. [11] Institute for Community Medicine, Study of Health in Pomerania/KEF, University Medicine Greifswald, 17475 Greifswald, Germany. [12] Institute of Human Genetics and Department of Genomics, Life & Brain Center, University of Bonn, 53127 Bonn, Germany. Ingo Marenholz and Sarah Grosche contributed equally to this work. Kirsten Beyer and Young-Ae Lee jointly supervised this work. Correspondence and requests for materials should be addressed to Y.-A.L. (email: yolee@mdc-berlin.de)

Food allergy (FA) is a worldwide increasing public health problem. Disease prevalence ranges from 1 to 5% in European countries to over 10% in Australia and most studies have demonstrated a consistent increase over the past two decades. In infancy, allergic responses against hen's egg (HE) and cow's milk (CM) are most common whereas in childhood, peanut (PN) allergy becomes more frequent[1–4]. Affected children can experience severe allergic reactions rendering food allergy the most common cause of life-threatening anaphylaxis in childhood[5–7]. Food allergy is a complex disease with genetic and environmental factors involved. Twin studies estimated food allergy heritability at about 80%[8, 9]. Previous genetic studies mainly focused on PN allergy. They pointed to a few genes which were repeatedly associated with the disease. Loss-of-function (LOF) mutations in the epidermal barrier gene filaggrin (FLG) increased the risk for PN sensitization and PN allergy, likely due to increased allergen penetration and sensitization to PN through a defective skin barrier[10]. In addition, association of the HLA-DQB1 locus with PN allergy has been reported by several independent groups[11–13]. The first genome-wide association study (GWAS) defined food allergy based on a convincing history of an allergic reaction to a specific food and evidence of sensitization to the same food. This study confirmed the HLA locus on chromosome 6 to be associated with PN allergy[14]. Although a skin barrier defect and immunological dysfunction seem to play a role in the development of food allergy, additional genetic factors remain to be identified.

Many studies on food allergy suffer from weak phenotype definitions as they rely on individual or parental reports of an adverse reaction to food. However, due to the large spectrum of disease symptoms which may affect any organ system, a reliable diagnosis is often difficult to obtain. A systematic review of the literature revealed that the point prevalence of self-reported food allergy was approximately six times higher than the point prevalence of challenge-proven food allergy[3], suggesting that over 80% of history-based food allergy diagnoses cannot be confirmed in an oral food challenge (OFC). Even if the diagnosis was based on reported allergic symptoms plus elevated specific IgE or a positive skin prick test, the prevalence of food allergy was overestimated by a factor of 3 compared with challenge-proven food allergy[3]. As a consequence, and in order to standardize the diagnostic criteria of food allergy, current guidelines recommend oral food challenges as the diagnostic gold standard for food allergy[3, 15].

Here, we report a GWAS on OFC-proven food allergy in a German study population. The discovery set includes 523 food allergic children and 2682 population-based controls. We replicate the results in a second data set of 380 German cases and 986 controls. As additional replication set the Chicago Food Allergy Study, comprising 671 FA cases and 1526 mostly family based controls, is available[14]. We identify five genomic regions to be associated with food allergy at genome-wide significance, of which the SERPINB gene cluster has not been linked to allergic disease previously. The identified food allergy susceptibility loci support the role of epithelial barriers and of the immune response in the development of food allergy.

## Results

**Study design and quality control**. To identify genes involved in food allergy, we performed a GWAS in a discovery set of 523 food allergic cases and 2682 population-based controls from the German Heinz Nixdorff Recall Study (HNR)[16]. All cases were from the Genetics of Food Allergy Study (GOFA) in which the diagnosis of food allergy was based on OFCs according to current guidelines[3]. After applying stringent quality control criteria (see

"Methods"), the discovery set consisted of 497 cases and 2387 controls with high quality genotype data. For imputation, we used the Haplotype Reference Consortium data at the University of Michigan as reference[17]. In order to ascertain high quality of imputed genotypes, we filtered for a quality score of $r^2 > 0.5$ and excluded low frequency variants (minor allele frequency (MAF) <5%), yielding over five million single nucleotide polymorphisms (SNPs) available for analysis. The study design is summarized in Fig. 1. Association with disease was calculated with FastLMM using an additive allele-dosage model. The main analysis was performed on any food allergy (Fig. 2). In addition, we investigated the three most common allergies against specific foods separately, including HE ($n = 288$), PN ($n = 220$), and CM ($n = 169$; Fig. 2). There was no evidence for inflation of the test statistics neither in the GWAS on any food allergy ($\lambda = 1.03$) nor in the allergen-stratified analyses (Supplementary Fig. 1). For replication, we considered all loci with moderate association in the discovery set ($P < 1 \times 10^{-3}$; Supplementary Data 1a–d). If multiple SNPs from the same locus reached the selection threshold, we selected the best SNP. For the remaining SNPs, we calculated linkage disequilibrium (LD) with the lead variant. If

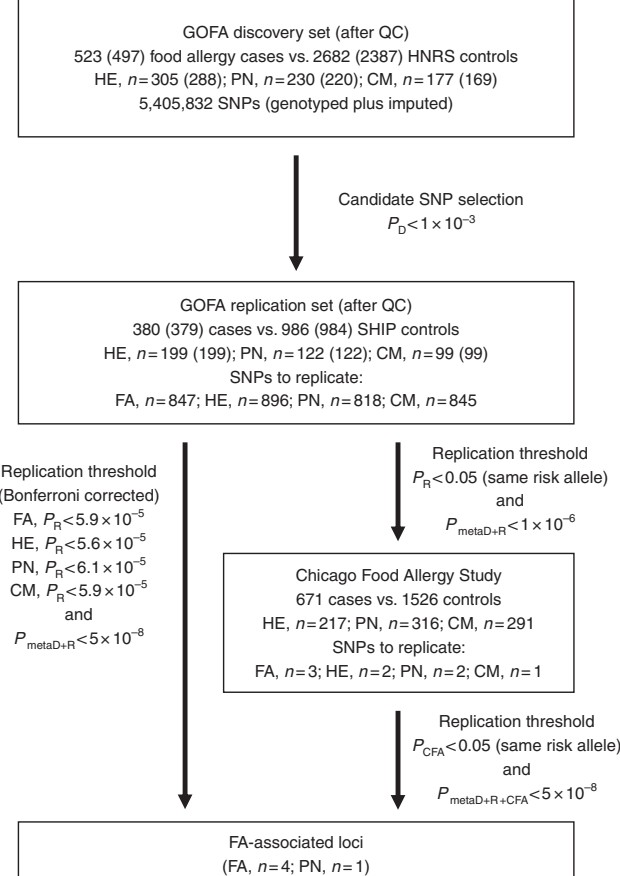

**Fig. 1** Study design of the GWAS on food allergy. Workflow of the GWAS on food allergy is shown including the number of cases and controls, and the P value thresholds used for each study set and phenotype. *GOFA* German Genetics of Food Allergy Study, *HNR* Heinz Nixdorf Recall Study, *SHIP* Study of Health in Pomerania, *CFA* Chicago Food Allergy Study, *QC* quality control, *FA* any food allergy, *HE* hen's egg allergy, *PN* peanut allergy, *CM* cow's milk allergy, $P_D$, $P_R$, and $P_{metaD+R}$, P value in the GOFA discovery set, in the GOFA replication set, and in the meta-analysis of both sets, respectively

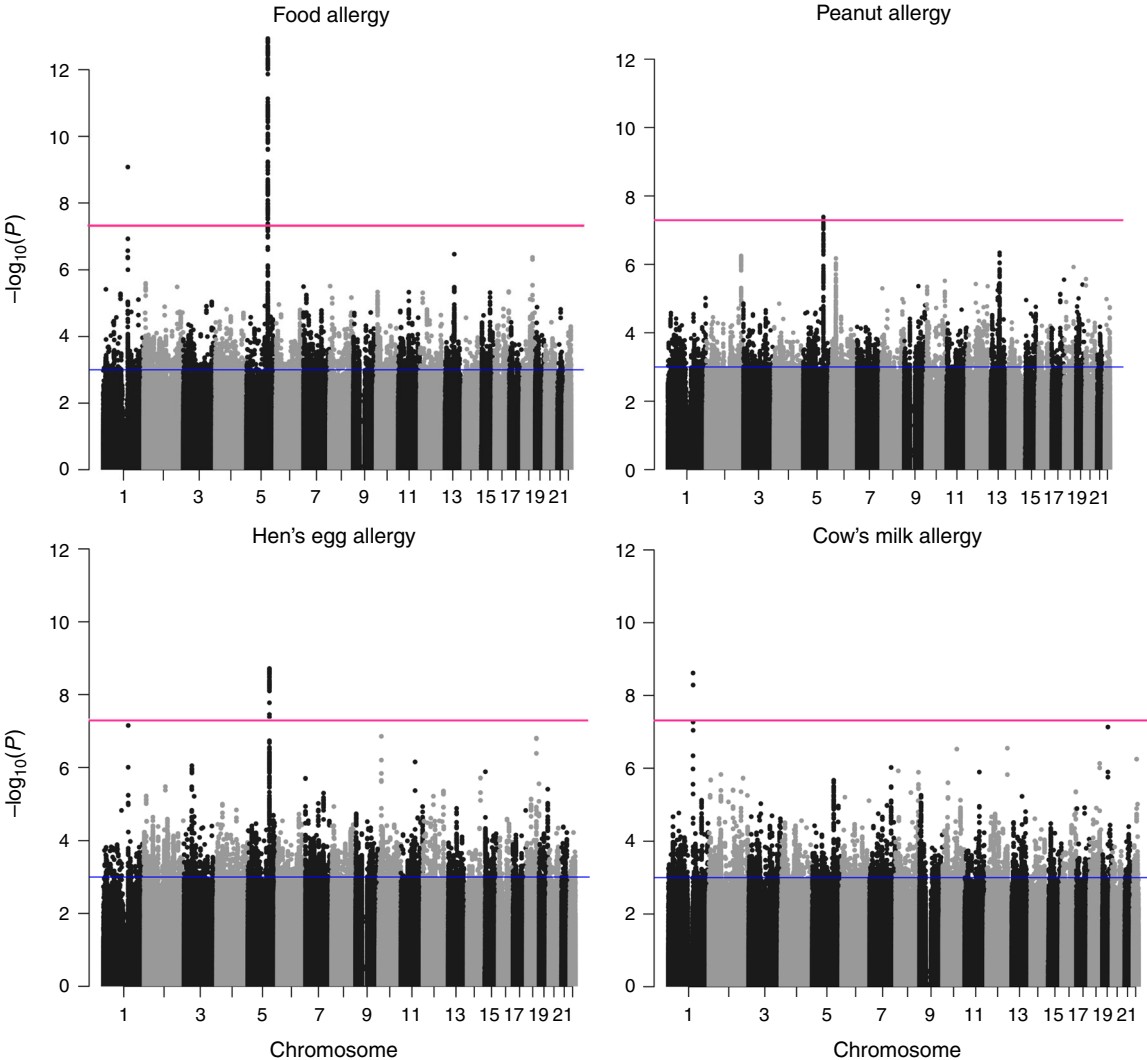

**Fig. 2** Association results of the GWAS on food allergy. The results of the GWAS on any food allergy (497 cases vs. 2387 controls) and on food-specific allergies against hen's egg (288 cases), peanut (220 cases), and cow's milk (169 cases) are presented. The Manhattan plots show the chromosomal positions (x-axis) and association P values (y-axis) for all SNPs of the discovery set. Red and blue lines indicate the thresholds for genome-wide significance ($P < 5 \times 10^{-8}$) and for entering the replication phase ($P < 1 \times 10^{-3}$) respectively

SNPs in low LD ($r^2 < 0.2$) were present, again the best SNP was selected for replication.

To replicate the findings, we investigated 380 additional food allergic cases of the GOFA study and 986 population-based controls of the Study of Health in Pomerania (SHIP)[18], of which 379 and 984 samples passed the quality check. All individuals of the discovery and replication set were of European ancestry as confirmed by principal component analysis. A detailed characterization of both data sets is provided in Table 1.

Furthermore, variants replicating with the same risk allele as the GOFA discovery set at nominal significance ($P < 0.05$) and not reaching the Bonferroni corrected P value in the GOFA replication set were confirmed in the Chicago Food Allergy Study (Supplementary Table 1) comprising 671 food allergic children, 144 non-allergic, non-sensitized normal controls, and 1382 controls of unknown phenotype (234 children and 1148 parents), all of European ancestry (Fig. 1)[14].

**Loci associated with food allergy**. In the main analysis on any food allergy, two loci already showed association at genome-wide

**Table 1 Characterization of the German Genetics of Food Allergy Study (GOFA)**

| | GOFA discovery set | GOFA replication set |
|---|---|---|
| *n* | 523 | 380 |
| Sex | 63% males | 66% males |
| Mean age | 2.1 years | 2.8 years |
| *Food allergies* | | |
| Hen's egg | 305 (58%) | 199 (52%) |
| Peanut | 230 (44%) | 122 (32%) |
| Cow's milk | 177 (34%) | 99 (26%) |

significance ($P < 5 \times 10^{-8}$) in the discovery set (Table 2, Fig. 2). The respective lead SNPs, rs12123821 on chromosome 1q21.3 (odds ratio (OR), 2.55; $P = 8.4 \times 10^{-10}$) and rs11949166 on 5q31.1 (OR, 0.60; $P = 1.2 \times 10^{-13}$), were also significantly associated with food allergy in the GOFA replication set after correction for the number of tests performed (for FA; $n = 847$, $P < 5.9 \times 10^{-5}$, Bonferroni correction). They replicated with the same risk alleles

**Table 2 Association of the identified susceptibility loci with different food allergies in the GOFA discovery and replication sets**

| SNP ID | Chr | Position[a] | EA/AA | Trait | GOFA discovery set | | | GOFA replication set | | | Meta-analysis | | |
|---|---|---|---|---|---|---|---|---|---|---|---|---|---|
| | | | | | AF | OR | P value | AF | OR | P value | AF | OR | P value |
| rs12123821 | 1 | 152179152 | T/C | **FA** | 0.058 | **2.55** | **$8.4 \times 10^{-10}$** | 0.060 | **2.86** | **$6.1 \times 10^{-7}$** | 0.058 | **2.65** | **$2.6 \times 10^{-15}$** |
| *FLG-AS1* (intron_variant) | | | | PN | | 2.35 | $1.5 \times 10^{-4}$ | | 3.50 | $8.3 \times 10^{-6}$ | | 2.69 | $1.3 \times 10^{-8}$ |
| | | | | HE | | 2.67 | $7.0 \times 10^{-8}$ | | 2.89 | $1.7 \times 10^{-5}$ | | 2.74 | $5.4 \times 10^{-12}$ |
| | | | | CM | | 3.59 | $2.4 \times 10^{-9}$ | | 3.24 | $4.1 \times 10^{-4}$ | | 3.49 | $5.2 \times 10^{-12}$ |
| rs11949166 | 5 | 132027681 | T/A | **FA** | 0.719 | **0.60** | **$1.2 \times 10^{-13}$** | 0.696 | **0.69** | **$3.0 \times 10^{-5}$** | 0.712 | **0.63** | **$4.3 \times 10^{-17}$** |
| *IL4/KIF3A* (intergenic) | | | | PN | | 0.63 | $4.3 \times 10^{-6}$ | | 0.72 | $2.3 \times 10^{-3}$ | | 0.66 | $3.6 \times 10^{-8}$ |
| | | | | HE | | 0.60 | $2.0 \times 10^{-9}$ | | 0.66 | $1.9 \times 10^{-4}$ | | 0.62 | $2.2 \times 10^{-12}$ |
| | | | | CM | | 0.64 | $1.9 \times 10^{-5}$ | | 0.79 | 0.054 | | 0.68 | $4.6 \times 10^{-6}$ |
| rs9273440 | 6 | 32627561 | C/T | FA | 0.706 | 0.98 | 0.053 | 0.707 | 0.75 | 0.088 | 0.706 | 0.89 | 0.010 |
| *HLA-DQB1* (3-prime UTR variant) | | | | **PN** | | **0.66** | **$6.6 \times 10^{-7}$** | | **0.45** | **$3.8 \times 10^{-6}$** | | **0.58** | **$1.7 \times 10^{-11}$** |
| | | | | HE | | 1.35 | 0.61 | | 0.90 | 0.65 | | 1.17 | 0.90 |
| | | | | CM | | 1.15 | 0.47 | | 0.89 | 0.78 | | 1.05 | 0.45 |
| rs2212434 | 11 | 76281593 | T/C | **FA** | 0.459 | **1.29** | **$3.4 \times 10^{-4}$** | 0.477 | **1.47** | **$8.2 \times 10^{-5}$** | 0.465 | **1.35** | **$1.4 \times 10^{-7}$** |
| *C11orf30/LRRC32* | | | | PN | | 1.25 | 0.020 | | 1.65 | $4.2 \times 10^{-3}$ | | 1.36 | $7.2 \times 10^{-6}$ |
| (intergenic_variant) | | | | | | | | | | | | | |
| | | | | HE | | 1.37 | $2.3 \times 10^{-4}$ | | 1.43 | 0.012 | | 1.39 | $8.8 \times 10^{-6}$ |
| | | | | CM | | 1.37 | $3.3 \times 10^{-3}$ | | 1.39 | 0.039 | | 1.38 | $3.4 \times 10^{-4}$ |
| rs12964116 | 18 | 61442619 | G/A | **FA** | 0.050 | **1.90** | **$5.7 \times 10^{-6}$** | 0.051 | **1.69** | **$9.4 \times 10^{-3}$** | 0.050 | **1.82** | **$2.4 \times 10^{-7}$** |
| *SERPINB7* (intron variant) | | | | PN | | 2.42 | $1.2 \times 10^{-6}$ | | 1.44 | 0.35 | | 2.11 | $7.2 \times 10^{-6}$ |
| | | | | HE | | 1.48 | 0.032 | | 1.82 | 0.019 | | 1.60 | $1.7 \times 10^{-3}$ |
| | | | | CM | | 1.77 | $2.2 \times 10^{-3}$ | | 1.71 | 0.18 | | 1.75 | $1.1 \times 10^{-3}$ |
| rs1243064 | 18 | 61513975 | A/T | FA | 0.259 | 1.48 | $4.3 \times 10^{-7}$ | 0.253 | 1.07 | 0.20 | 0.257 | 1.31 | $2.1 \times 10^{-6}$ |
| *SERPINB7/B2* (intergenic_variant) | | | | PN | | 1.49 | $4.6 \times 10^{-5}$ | | 1.07 | 0.38 | | 1.34 | $1.4 \times 10^{-4}$ |
| | | | | **HE** | | **1.65** | **$1.6 \times 10^{-7}$** | | **1.21** | **0.028** | | **1.47** | **$4.2 \times 10^{-8}$** |
| | | | | CM | | 1.80 | $9.8 \times 10^{-7}$ | | 1.03 | 0.36 | | 1.51 | $7.5 \times 10^{-6}$ |

AA alternative allele, AF effect allele frequency, CM cow's milk, EA effect allele, FA food allergy, GOFA Genetics of Food Allergy Study, OR odds ratio, HE hen's egg, PN peanut. Strongest association for each SNP is indicated in bold
[a]Genomic positions were based on human genome reference assembly GRCh37.p13

and with similar effect sizes (rs12123821; OR, 2.86; $P = 6.1 \times 10^{-7}$ and rs11949166; OR, 0.69; $P = 3.0 \times 10^{-5}$). Meta-analysis of the two sets yielded highly significant associations at 1q21.3 and 5q31.1 (rs12123821; OR, 2.65; $P = 2.6 \times 10^{-15}$ and rs11949166; OR, 0.63; $P = 4.3 \times 10^{-17}$).

Variant rs12123821 at 1q21.3 is located within the epidermal differentiation complex (EDC) near the epidermal barrier gene filaggrin (*FLG*, Supplementary Fig. 2a) which was previously associated with PN allergy[10]. Since we identified LD between rs12123821 and a LOF mutation in *FLG*, c.2282del4 ($r^2 = 0.19$, $D' = 0.78$), we evaluated whether the association signal at 1q21.3 was due to known *FLG* mutations. We included the two most common *FLG* LOF mutations in European populations, *FLG* c.2282del4 (tagged by rs12123821) and p.R501X (rs61816761) as covariates in the analysis which eliminated the highest association peaks within the EDC (Supplementary Table 2). While our results confirmed the role of *FLG* null mutations in food allergy, a residual association was still detectable between *FLG* and the repetin gene (*RPTN;* Supplementary Fig. 2b, Supplementary Table 2), which could point to additional genetic risk factors in this region.

*FLG* mutations are known to be strong risk factors for eczema[19] which often co-occurs with food allergy[4]. In order to exclude that the observed association was due to an underlying association with eczema, we performed the association analysis in the subset of children without eczema ($n = 152$, Table 3). The effect of rs12123821 remained significant with similar effect size (OR 1.77; 95% CI 1.15–2.74; $P = 0.0094$), demonstrating an eczema-independent effect of *FLG* null mutations on food allergy. Finally, we investigated the effect of *FLG* null mutations on allergies to specific foods. While the association of *FLG* null mutations with PN allergy is well-documented[10], we show that

*FLG* mutations also confer risk for HE and CM allergy with similar and large effect sizes (Table 2).

On chromosome 5q31.1, the strongest association was observed for rs11949166 located between the interleukin 4 gene (*IL4*) and the kinesin family member 3a gene (*KIF3A*) within the cytokine gene cluster (Table 2). Variants spanning the whole 0.2 Mb region from *IL5* to *KIF3A* were associated with food allergy at genome-wide significance ($P < 5 \times 10^{-8}$, Supplementary Fig. 3a). We tested whether LD within the cytokine gene cluster accounted for the multitude of associated SNPs or whether several independent signals were present. We identified two groups of SNPs covering *IL5/RAD50* and *IL4/KIF3A*, which were significantly associated with food allergy (Supplementary Figs. 3a and 4a). There was high LD between the SNPs of each group, but low LD between SNPs of different groups. Mutual adjustment for the lead SNP of each group pointed to two independent signals (Supplementary Table 3, Supplementary Figs. 3b and 4b). Since this chromosomal region is a known eczema locus, we again stratified the association analysis for the eczema status and confirmed an eczema-independent effect of rs11949166 on food allergy with nearly identical effect sizes in the subgroups with eczema (OR, 1.69; 95% CI, 1.50–1.91) and without eczema (OR, 1.61; 95% CI, 1.27–2.04; Table 3).

Two novel susceptibility loci were identified at genome-wide significance after replication in the Chicago Food Allergy Study (Supplementary Table 1). The lead variant at 11q13.5, rs2212434 (Supplementary Fig. 5), was consistently associated with the same risk allele in all three study populations (Table 4). The same SNP was identified as the best associated variant at this locus in the largest eczema GWAS[20] to date. The eczema-stratified analysis revealed a strong and significant effect (OR, 1.40; 95% CI, 1.25–1.58; $P = 1.9 \times 10^{-8}$) in the food allergy plus eczema group

**Table 3 Risk estimates for the identified food allergy loci dependent on the eczema status**

| SNP ID | Chr | Cases (n) | Controls (n) | EA | AA | OR | 95% CI | P value |
|---|---|---|---|---|---|---|---|---|
| *Food allergy, no eczema* | | | | | | | | |
| rs12123821 | 1 | 152 | 3373 | T | C | 1.77 | 1.15–2.74 | $9.4 \times 10^{-3}$ |
| rs11949166 | 5 | 152 | 3373 | A | T | 1.61 | 1.27–2.04 | $8.9 \times 10^{-5}$ |
| rs9273440 | 6 | 71 (PN) | 3373 | T | C | 1.67 | 1.19–2.35 | $3.2 \times 10^{-3}$ |
| rs2212434 | 11 | 152 | 3373 | T | C | 1.14 | 0.90–1.44 | 0.29 |
| rs12964116 | 18 | 152 | 3373 | G | A | 2.26 | 1.49–3.43 | $1.3 \times 10^{-4}$ |
| *Food allergy plus eczema* | | | | | | | | |
| rs12123821 | 1 | 717 | 3373 | T | C | 2.13 | 1.74–2.62 | $5.6 \times 10^{-15}$ |
| rs11949166 | 5 | 717 | 3373 | A | T | 1.69 | 1.50–1.91 | $2.4 \times 10^{-17}$ |
| rs9273440 | 6 | 265 (PN) | 3373 | T | C | 1.64 | 1.36–1.97 | $2.0 \times 10^{-7}$ |
| rs2212434 | 11 | 717 | 3373 | T | C | 1.40 | 1.25–1.58 | $1.9 \times 10^{-8}$ |
| rs12964116 | 18 | 717 | 3373 | G | A | 1.78 | 1.41–2.25 | $1.2 \times 10^{-6}$ |

AA alternative allele, AF effect allele frequency, CI confidence interval, EA effect allele, OR odds ratio, PN peanut allergy

and a residual effect (OR, 1.14; 95% CI, 0.90–1.44; $P = 0.29$) that was, however, not significant in a small set of 152 food allergic children without eczema (Table 3).

Another new susceptibility locus, which had not yet been linked to any allergic disease, was identified in chromosomal region 18q21.3 (Supplementary Fig. 6a). SNP rs12964116 located in intron 1 of *SERPINB7* (serpin peptidase inhibitor, clade B, member 7) was associated with food allergy in the GOFA discovery set (OR, 1.9; $P = 5.7 \times 10^{-6}$, Table 2) and replicated with the same risk allele and a similar effect size in the GOFA replication set (OR, 1.69; $P = 9.4 \times 10^{-3}$). Since rs12964116 did not reach the Bonferroni corrected $P$ value in the replication set ($P < 5.9 \times 10^{-5}$), we investigated the Chicago Food Allergy Study[14] in order to confirm this locus. Again, the SNP was significantly associated with food allergy and with PN allergy (Table 4), reaching genome-wide significance for both phenotypes in the meta-analysis including all three studies ($P = 1.8 \times 10^{-8}$ for any food allergy and $P = 1.9 \times 10^{-10}$ for PN allergy). Within the SERPINB gene cluster, a second SNP (rs1243064) in moderate LD with rs12964116 ($r^2 = 0.06$, $D' = 0.71$) was associated with food allergy (Supplementary Fig. 7a). In order to explore whether the two SNPs represented independent association signals we mutually conditioned on the two lead variants (Supplementary Table 5). In both cases, association of the other variant with food allergy decreased but was still present suggesting more than one risk haplotype at this locus (Supplementary Figs. 6b and 7b, Supplementary Table 5).

Association of rs1243064 was confirmed in the GOFA replication set for HE allergy at nominal significance, reaching genome-wide significance in the meta-analysis of GOFA discovery and replication set ($P = 4.2 \times 10^{-8}$, Table 2). In the Chicago Food Allergy Study the same risk allele was identified (Table 4). However, association did not reach significance ($P = 0.15$) which may be due to reduced power in a small sample with a less stringent phenotype definition that was not based on OFCs. Overall, association at the serpin locus was consistent and strong for any food allergy, PN, and HE allergy.

To better understand the potential functional basis of the novel food allergy locus, we used LDLink[21] to identify all variants within the SERPINB gene cluster which are in high LD ($r^2 > 0.8$) with the two lead SNPs, rs12964116 and rs1243064. None of the identified candidate SNPs altered any protein sequences as predicted by the ENSEMBL variant effect predictor (Supplementary Table 6)[22]. We therefore evaluated their association with gene expression in expression databases including the Genotype-Expression database (GTEx, version V6p), and reviewed their functional annotations in the ENCODE Consortium (http://genome.ucsc.edu/ENCODE/)[23].

rs12964116 is located in an intron of *SERPINB7* in a binding site for several members of the transcription factor activator protein (AP)-1 complex, which is involved in diverse cellular processes including cell growth and differentiation (Supplementary Table 6). In chromatin immunoprecipitation (ChIP)-seq experiments this site has also been shown to bind the transcription factor CCAAT/enhancer-binding protein beta (CEBPB)[24] which regulates the expression of genes involved in immune and inflammatory responses, including cytokines interleukin-6, interleukin-4, interleukin-5, and TNF-alpha, as well as signal transducer and activator of transcription 3 (STAT3) which mediates the transcriptional activation in response to multiple cytokines and growth factors. The other lead SNP, rs1243064, is a tissue-specific expression quantitative trait locus (eQTL), with the risk allele rs1243064A being negatively correlated with *SERPINB10* expression in whole blood (Supplementary Table 6).

We then used LD score regression analysis in order to quantify the liability-scale heritability of food allergy that was explained by the lead variants identified in our study. Altogether, the food allergy susceptibility loci identified in this study explained ~10.2% of the variance in liability (Supplementary Table 7).

**Loci associated with allergy to specific foods.** Association results within the HLA region at 6p21 (Supplementary Fig. 8a) confirmed a previously reported locus for PN allergy[14]. We found LD ($r^2 = 0.48$, $D' = 0.85$) between rs9273440, which was strongly associated with PN allergy in the discovery and in the replication set (Table 2), and rs9275596, which was the lead SNP in the previous GWAS[14]. Although several SNPs reached the selection threshold ($P < 1 \times 10^{-3}$) in the discovery set, conditioning on rs9273440 eliminated all association signals within the region (Supplementary Fig. 8b), pointing to a single signal at this locus. Notably, children with HE or CM allergy did not contribute to the association with rs9273440 (Table 2) demonstrating a PN-specific locus.

In the analysis of HE allergy, 896 candidate variants were identified in the discovery set (Supplementary Data 1b). Apart from the susceptibility loci for any food allergy at 1q21.3 and 5q31.1 (Table 2), two additional SNPs were significantly associated in the GOFA replication set and selected for replication in the Chicago Food Allergy Study in which neither SNP reached significance (Supplementary Table 1). In the analysis of CM allergy, 845 SNPs were selected for replication (Supplementary Data 1d). One candidate SNP specific for CM allergy (rs73908987) replicated in the GOFA replication set (Supplementary Table 1) reaching $6.0 \times 10^{-7}$ in the meta-analysis of the

**Table 4 Replication of *C11orf30/LRRC32* (rs2212434) and the clade B serpins gene cluster (rs12964116 and rs1243064) in the Chicago Food Allergy Study**

| SNP ID | EA/AA | Trait | GOFA discovery set | | | GOFA replication set | | | Chicago Food Allergy Study | | | Meta-analysis |
|--------|-------|-------|--------------------|--|--|----------------------|--|--|----------------------------|--|--|----------------|
| | | | $AF_{cases}$ | $AF_{controls}$ | *P* value | $AF_{cases}$ | $AF_{controls}$ | *P* value | $AF_{cases}$ | $AF_{controls}^a$ | *P* value | *P* value |
| rs2212434 | T/C | FA | 0.511 | 0.448 | $3.4 \times 10^{-4}$ | 0.540 | 0.451 | $8.2 \times 10^{-5}$ | 0.512 | 0.469 | $1.4 \times 10^{-4}$ | $9.2 \times 10^{-11}$ |
| | | HE | 0.528 | 0.448 | $2.3 \times 10^{-4}$ | 0.525 | 0.451 | 0.012 | 0.492 | 0.471 | 0.24 | $1.3 \times 10^{-5}$ |
| | | PN | 0.505 | 0.448 | 0.020 | 0.550 | 0.451 | $4.2 \times 10^{-3}$ | 0.497 | 0.471 | 0.12 | $2.3 \times 10^{-4}$ |
| | | CM | 0.530 | 0.448 | $3.3 \times 10^{-3}$ | 0.530 | 0.451 | 0.039 | 0.514 | 0.469 | $9.8 \times 10^{-3}$ | $1.3 \times 10^{-5}$ |
| rs12964116 | G/A | FA | 0.080 | 0.043 | $5.7 \times 10^{-6}$ | 0.069 | 0.044 | $9.4 \times 10^{-3}$ | 0.061 | 0.050 | 0.010 | $1.8 \times 10^{-8}$ |
| | | HE | 0.063 | 0.043 | 0.32 | 0.070 | 0.044 | 0.019 | 0.065 | 0.052 | 0.14 | $6.0 \times 10^{-4}$ |
| | | PN | 0.095 | 0.043 | $1.2 \times 10^{-6}$ | 0.058 | 0.044 | 0.35 | 0.084 | 0.050 | $5.8 \times 10^{-6}$ | $1.9 \times 10^{-10}$ |
| | | CM | 0.077 | 0.043 | $2.2 \times 10^{-3}$ | 0.061 | 0.044 | 0.18 | 0.041 | 0.051 | 0.29 | 0.11 |
| rs1243064 | A/T | FA | 0.322 | 0.246 | $4.3 \times 10^{-7}$ | 0.278 | 0.244 | 0.20 | 0.265 | 0.264 | 0.56 | $5.1 \times 10^{-5}$ |
| | | HE | 0.343 | 0.246 | $1.6 \times 10^{-7}$ | 0.306 | 0.244 | 0.028 | 0.288 | 0.262 | 0.15 | $8.0 \times 10^{-8}$ |
| | | PN | 0.325 | 0.246 | $4.6 \times 10^{-5}$ | 0.267 | 0.244 | 0.38 | 0.279 | 0.264 | 0.20 | $2.2 \times 10^{-4}$ |
| | | CM | 0.358 | 0.246 | $9.8 \times 10^{-7}$ | 0.280 | 0.244 | 0.36 | 0.244 | 0.261 | 0.31 | 0.013 |

AA alternative allele, AF effect allele frequency, CM cow's milk, EA effect allele, FA food allergy, GOFA Genetics of Food Allergy Study, HE hen's egg, PN peanut
$^a$ Controls of the Chicago Food Allergy Study include unaffected controls ($n = 144$) and controls of unknown phenotype ($n = 1382$). The AFs in both groups are presented in Supplementary Table 4

two GOFA sets. Unfortunately, there were no data or proxy SNPs ($r^2 > 0.8$) available for this variant in the Chicago Food Allergy Study.

## Discussion

Here we report a GWAS on OFC-proven food allergy which stratified the results for the three most common food allergens. We identified five loci at genome-wide significance including the SERPINB gene cluster at 18q21.3 which was not previously linked to any allergic disease. We demonstrated that the association of *FLG* at 1q21.3, the cytokine gene cluster at 5q31.1, the *C11orf30/LRRC32* region on chromosome 11q13.5, and the SERPINB gene cluster with food allergy was independent of the allergen whereas the HLA locus at 6p21 was clearly identified as a PN allergy-specific susceptibility locus. Moreover, we showed that the effect on food allergy was independent of eczema for 4 out of 5 susceptibility loci.

Two recent GWAS focussed on PN allergy. The Chicago Food Allergy Study defined cases based on a reported history of food allergy in combination with elevated specific IgE against the same food and confirmed a previous association of the HLA region with PN allergy[14]. The HLA association with PN allergy was also confirmed in the HealthNuts Study. In the latter report, PN allergy was diagnosed by OFC but results did not reach genome-wide significance, likely due to the small sample size (73 cases and 148 controls)[25]. In the present study, the strongest association at 6p21 was observed for rs9273440 located in the 3′-untranslated region (UTR) of HLA-DQB1. This marker was in LD with the two variants reported by Hong et al.[14], rs9275596 ($r^2 = 0.48$, $D' = 0.85$) and rs7192 ($r^2 = 0.25$, $D' = 0.69$). The functional investigation revealed high LD with a missense variant in *HLD-DQB1*. Although this variant is located in a highly conserved region there is no additional evidence for a disease-causing role. Our study design enabled us to investigate the effect of HLA markers on food-specific allergies. Clearly, the lead SNP in the HLA region, rs9273440, was only associated with PN allergy and not with HE or CM allergy.

A strong association with food allergy, reaching genome-wide significance already in the discovery stage, was found in the EDC on chromosome 1. We showed that this association was attributable to null mutations in the filaggrin gene which encodes an epidermal barrier protein. Filaggrin null mutations are strong risk factors for eczema and eczema-associated asthma or allergic rhinitis[19, 26, 27]. In addition, an association of *FLG* mutations with

PN allergy has previously been reported[10]. Here, we demonstrate that *FLG* mutations have an effect on food allergy regardless of the causing allergen with consistently large effect sizes for HE, PN, and CM (Table 2). Furthermore, and in contrast to a previous study that reported an association of *FLG* mutations with food allergy only in the context of eczema[28], our large study population enabled us to compare the *FLG* effect between food allergic children with and without eczema. Although the effect size in children with eczema was moderately larger (OR = 2.13, 95% CI 1.74–2.62), we clearly observed a significant, strong effect of *FLG* mutations on food allergy in the absence of eczema (OR 1.77, 95% CI 1.15–2.74). This finding is distinct from asthma for which a *FLG* effect was only detectable in the presence of eczema[19, 26, 27]. Interestingly, filaggrin is expressed in the oral and esophageal mucosa, but not in airway epithelia[29, 30]. While in eczema-associated asthma allergic sensitization through the defective skin barrier seems to play a pivotal role in disease development, in food allergy, enhanced penetration of allergens may occur through a leaky epithelial barrier in the upper gastrointestinal tract independently of the skin. A recent study suggested a link between downregulation of filaggrin in the esophageal mucosa and impairment of the corresponding epithelial barrier[30].

Association with food allergy was also observed within the cytokine gene cluster on chromosome 5q31.1, spanning the whole region from *IL5* to *KIF3A*. This region has previously been linked to a number of inflammatory and immune-related diseases including Crohn's disease[31], psoriasis[32], and eczema[33]. Since food allergy is often associated with eczema, we demonstrated that the observed association was independent of eczema. In both subgroups, food allergic children with and without eczema, rs11949166 showed highly significant association with nearly identical effect sizes. The lead SNP at this locus, rs11949166, was located between *IL4* and *KIF3A*. Using conditional analysis we clearly identify a second independent association signal in the *RAD50/IL13* region which also contains the well-known coding IL13 variant (IL-13 R130Q) involved in allergic disease[34]. Recent studies provided evidence that IL-4/IL-13 pathways play an important role in food allergy. Mouse models have demonstrated that IL-4 and IL-13 production by group 2 innate lymphoid cells (ILC2s) blocked the generation of mucosal allergen-specific regulatory T cells and promoted food allergy[35].

The *C11orf30/LRRC32* region is a known risk locus for eczema[36], asthma[37, 38], and other inflammatory diseases such as

inflammatory bowel disease[39]. The lead SNP rs2212434 has previously been identified in the largest meta-GWAS on eczema, in which the odds ratio was 1.09 (95% CI, 1.07–1.12)[20]. For food allergy, we estimated an OR of 1.35 (95% CI, 1.20–1.51) which was even higher when considering the combined food allergy plus eczema phenotype (OR, 1.40; 95% CI, 1.25–1.58). Since *C11orf30/LRRC32* was also strongly associated with the atopic march (rs2155219, OR, 1.33; 95% CI, 1.24–1.43)[40], our results support a key role of this locus in the development of multiple allergic disorders including food allergy.

Importantly, we report a novel, genome-wide significant association of food allergy with the SERPINB gene cluster on chromosome 18q21.3. Two SNPs in moderate LD were found to be associated with disease. One lead SNP, rs12964116, is located in an intron of *SERPINB7* which belongs to the serine protease inhibitor (serpin) superfamily. *SERPINB7* shows a very specific expression pattern. It is highly expressed in the upper layers of the epidermis. Loss-of-function mutations in *SERPINB7* cause "Nagashima-type" palmoplantar keratosis (NPPK), an autosomal recessive hyperkeratosis of the palms and soles which is associated with skin barrier deficiency[41]. Apart from the skin, *SERPINB7* is expressed in few other organs lined by stratified squamous epithelia including the esophagus (Supplementary Table 8). Its expression there suggests a potential role in the epithelial integrity and function of the upper digestive tract that may be relevant to the development of food allergy. rs12964116 alters a binding site for several transcription factors, including CEBPB and STAT3. *CEBPB* regulates the expression of genes involved in allergic inflammation, including the $T_h2$ cell (type 2 helper T cells) effector cytokines interleukin-13[42], interleukin-4,[43] and interleukin-5[44], and also promotes mucosal immunity in a mouse model of oropharyngeal candidiasis[45]. Furthermore, transcription factor STAT3 binds to this locus which is required for $T_h2$ cell development in a mouse model of allergic inflammation[46]. The other lead SNP, rs1243064, is a tissue-specific eQTL, with the risk allele rs1243064A being negatively correlated with the expression of *SERPINB10* in whole blood. *SERPINB10* is expressed in leukocytes, blood, esophagus, and skin, but little is known about its function.

Although the functional data point to *SERPINB7* and *SERPINB10*, other SERPINB genes also represent good candidates as they constitute a tightly linked gene cluster on chromosome 18. Clade B serpins are involved in several biological functions including protease inhibition, tumor suppression, regulation of apoptosis and inflammation. Of note, many clade B serpins have very restricted expression patterns with high expression levels in the esophageal mucosa (Supplementary Table 8).

SerpinB3 and serpinB4 were upregulated in affected skin of eczema patients and in the airway epithelia of patients with allergic asthma, induced via a pathway involving the $T_h2$ cytokines IL-4 and IL-13[47–49]. In a mouse model, allergen exposure induced *Serpinb3a* expression in the epidermis which promoted barrier dysfunction and skin inflammation[50]. In contrast, Serpinb2 had a protective effect on the skin barrier; a knockout mouse revealed loss of stratum corneum integrity and reduced barrier function[51]. SerpinB2 was upregulated in airway epithelial cells from asthmatic patients[52] and upon allergen challenge[53]. After enteric nematode infection, IL-4 and IL-13 induced *SERPINB2* expression in the intestinal mucosa where it affects diverse immunological processes; serpinB2 protects macrophages from apoptosis and is involved in the regulation of cytokines[54]. Functional studies will be required to gain a better understanding of the physiological role of clade B serpins in food allergy.

We estimated the FA heritability from our GWAS summary statistics to be 24.4%. This is in contrast to previous estimates from twin studies which yielded estimates around 80%. Similar discrepancies have been reported for other complex diseases[55]. The difference between the heritability estimates from GWAS and from pedigree or twin studies may be due to an underestimation of the contribution of common environmental factors in twin studies, gene-environment interactions or model misspecification[56]. The five lead variants of the food allergy susceptibility loci identified in this study explained 10.2% of the heritability.

Some limitations of our study need to be discussed. Although we performed the largest GWAS on food allergy to date, the sample size was relatively small for a GWAS on a complex genetic trait. This was due to the strict phenotype definition used in our study. Though recommended in current guidelines[3, 15], OFCs are not always performed as the diagnostic standard method. As a consequence for challenge-proven food allergy, large numbers of patients are difficult to obtain. This study was powered to detect loci of relatively large effect sizes. While the combined GOFA set had a power of 99% to detect a risk variant with an OR of 1.6 (using an allelic model with 5% prevalence, 20% risk allele frequency, and alpha $<5 \times 10^{-8}$)[57], the power dropped to 10% for variants with a moderate OR of 1.3. Additional studies in larger samples will be required to evaluate variants with low effect sizes.

In the context of limited study power, the finding of allergen-specific associations should be interpreted cautiously since the lack of association between an allergen-specific SNP and other FAs may be due to limited sample size of the subgroups. Likewise, our assessment of association with FA in the context of eczema was affected by sample size. In particular in the relatively small subset of patients without eczema ($n = 152$), a moderate SNP effect might not be detected. Accordingly, the effect of *C11orf30/LRRC32* on food allergy in the absence of eczema (OR, 1.14; 95% CI, 0.90–1.44, $p = 0.29$) might become significant, if larger study populations are analyzed. In contrast to the well phenotyped cases included in this study, there was no reliable information on food allergy available for the controls. Therefore, the presence of affected individuals among controls may have decreased the power of our study. Given a food allergy prevalence of about 5% in Western Europe, the loss of power was probably minor.

In summary, the identification of five food allergy susceptibility loci demonstrates that a strict phenotype definition, as set forth by recent medical guidelines on food allergy, is important for studies on the genetics of food allergy. The discovery of the SERPINB gene cluster in food allergy susceptibility emphasizes the importance of proteolytic pathways in the regulation of the immune response and in the maintenance of the epithelial barrier, both of which are impaired in food allergy.

## Methods

**Study populations and study design**. Children of the GOFA were recruited at clinical centers in Berlin, Wangen, Hannover, and Oldenburg, Germany. In total, 523 food allergic children were recruited for the GOFA discovery phase. Apart from any FA, we investigated the three most common food allergies against HE, PN, and CM. In total, 2682 German control individuals without information on food allergy originated from the HNR. HNR is a population-based cohort study for cardiovascular disease[16] comprising 4800 individuals from Ruhr area in Germany. The GOFA replication set comprised another 380 children with food allergy and 986 unphenotyped control individuals from the SHIP, a population-based cohort from North-Eastern Germany[18]. The second replication set, the Chicago Food Allergy Study, has previously been described in detail[14]. This study included 671 food allergic children of European ancestry of whom 316, 291, and 217 were allergic against PN, CM, and HE, respectively. In total, 144 non-allergic non-sensitized normal controls and 1382 individuals of unknown phenotype (234 children and 1148 parents) served as controls. This study has been approved by the ethics committees of Charité Universitätsmedizin Berlin, Hannover Medical School, and the Medical Associations of Berlin and Baden-Württemberg. The study protocol of the Chicago Food Allergy Study was approved by the Institutional Review Board of Ann and Robert H. Lurie Children's Hospital of Chicago and the Institutional Review Board of Johns Hopkins Bloomberg School of Public Health. Written informed consent has been given by all participants or their legal guardians.

For each phenotype under study (FA, HE, PN, and CM), all SNPs with moderate association in the discovery set ($P < 1 \times 10^{-3}$) were identified (FA 7699 SNPs, HE 8959 SNPs, PN 6794 SNPs, and CM 6955 SNPs). To define a locus, we grouped all consecutive SNPs with $P < 1 \times 10^{-3}$ and a distance <1 Mb to the next SNP (FA 611 loci, HE 634 loci, PN 595 loci, and CM 612 loci). At each locus, we selected the SNP with the lowest $P$ value as lead SNP. To identify additional, independent association signals within each locus, we identified all SNPs in low LD with the lead SNP ($r^2 < 0.2$) and again selected the best SNP. Thus the number of additional LD-selected SNPs was 236, 262, 223, and 233 SNPs, and the total number of candidate SNPs selected in the GOFA discovery set were 847, 896, 818, and 845 for the phenotypes FA, HE, PN, and CM, respectively.

In the GOFA replication set, significant association was defined as an association with the same risk allele as in the discovery set at the Bonferroni corrected $P$ value (0.05/number of SNPs tested for a given phenotype). The Bonferroni corrected significance thresholds were $P < 5.9 \times 10^{-5}$ for any FA, $P < 5.6 \times 10^{-5}$ for HE, $P < 6.1 \times 10^{-5}$ for PN, and $P < 5.9 \times 10^{-5}$ for CM. The threshold for genome-wide significance was $P < 5 \times 10^{-8}$ in the meta-analysis (Fig. 1).

Variants replicating at nominal significance ($P < 0.05$) in the GOFA replication set and yielding a $P < 10^{-6}$ in the meta-analysis of GOFA discovery and GOFA replication, but not reaching the Bonferroni corrected $P$ value, were additionally confirmed in the Chicago Food Allergy Study.

**Diagnostic criteria for food allergy**. According to the current guidelines, food allergy was diagnosed based on OFC ($n = 775$), most of which ($n = 650$, 83.9%) were conducted in a double-blind placebo controlled setting. Children with a convincing history of an immediate, severe allergic reaction plus specific sensitization to the same food (IgE > 0.35 kU l$^{-1}$) were included as cases without further challenge ($n = 127$), as OFC is contraindicated due to the risk of severe allergic reaction.

OFCs were performed in an inpatient hospital setting under physicians' supervision. Only one food or placebo was investigated per 24 h period and was administered in seven escalating doses at 30 min intervals. Consistent with PRACTALL guidelines, food challenges were scored as positive if objective cutaneous, gastrointestinal, respiratory or cardiovascular reactions attributable to the allergen, but not to placebo were observed[15].

**Diagnostic criteria for eczema**. A physician's diagnosis of eczema was made according to standard criteria in the presence of a chronic or chronically relapsing pruritic dermatitis with the typical morphology and distribution[58, 59]. More detailed information on eczema onset in the GOFA study sets is provided in Supplementary Note 1.

**Genotyping and quality control**. Samples were genotyped on Illumina's HumanOmniExpressExome-8 v1.2, HumanOmniExpress-12 v1.0, or HumanOmni1-Quad v1 (Supplementary Table 9). For the discovery and the replication set, the same QC criteria were applied. Individuals with a call rate <0.97 or with high heterozygosity (>0.35) were excluded. Individual SNPs were filtered according to the following criteria: (i) low call rate (<0.96 in cases or controls), (ii) low MAF <0.005 in cases or controls, (iii) genotypes out of Hardy–Weinberg equilibrium (HWE, $P < 0.00001$ in cases or $P < 0.0001$ in controls). SNPs with a call rate lower than 0.99 were excluded if having a MAF < 0.05 or if they were out of HWE ($P < 0.001$). Only SNPs fulfilling the above mentioned QC were used in subsequent steps. Genotypes of cases and controls were recoded to the "+" strand using the –flip command in PLINK[60]. Furthermore markers were deleted if allele frequencies in the HNR control population differed by more than 0.15 compared with the frequency in 379 Europeans available from the 1000 Genomes project. After quality control, the GOFA discovery and GOFA replication sets included 497 (before QC 523) and 379 (before QC 380) cases as well as 2387 (before QC 2682) and 984 (before QC 986) 1526 controls, respectively.

**Imputation**. Genotype imputation was performed separately for the discovery and the replication set on the Michigan Imputation Server (https://imputationserver.sph.umich.edu/index.html)[61] using the updated Haplotype Reference Consortium (version r1.1) panel[17]. Since genotyping was performed on different Illumina SNP arrays, we used only those SNPs for imputation which were genotyped on all arrays of a data set[62]. Cleaned SNP data were converted into vcf format and uploaded on the imputation server. For phasing SHAPEIT v2 was used[63], imputation was done with minimac3. After downloading the imputed data, we performed additional filtering steps; in both sets, we excluded SNPs with poor imputation quality ($r^2 < 0.5$), low allele frequency (MAF<5%), and deviation from HWE ($P < 10^{-12}$, in controls).

**Statistical analyses**. Association analysis and population stratification control was performed using FaST-LMM (v2.07) with imputed genotype dosages[64]. For each chromosome, a relationship matrix was computed using all autosomal, genotyped SNPs except those on the chromosome being analyzed. For conditional analyses, allele dosages of the corresponding SNPs were included as covariates. Odds ratios were obtained from association analysis with Mach2dat using logistic regression with sex and the first seven principal components as covariates. Association of the

lead SNPs with food allergy dependent on the eczema status was calculated by logistic regression with PLINK[60] using sex as covariate. The genomic inflation factor was calculated for each trait under study. $P$ values were meta-analyzed with METAL using the weighted $Z$-score method taking into account the sample size and the effect direction[65]. LD between SNPs in the control population of set 1 was calculated with PLINK[60]. Association analysis of the Chicago Food Allergy Study was performed using the modified quasi-likelihood score test under an additive genetic model.

**Functional annotation**. To identify potential functional variants within the food allergy loci, we used LD link[21] to identify all SNPs in high LD ($r^2 > 0.8$) with the lead SNPs. We subsequently used the Ensembl variant effect predictor[22], and the Genotype-Expression database (GTEx, V6p)[66] to review their functional annotations with respect to potential impact on protein structure, regulatory elements, tissue-specific gene expression, and transcription factor binding.

To detect associations of the food allergy loci with gene expression levels, we used the single nucleotide polymorphism annotator[67] tool to query publically available databases on expression quantitative trait loci (eQTL) in relevant tissues. If a SNP was associated with the expression of a gene in a tissue, we identified all independent eQTLs for that gene/tissue pair. To this end, we selected the best eQTL of the gene/tissue pair, then used LD link[21] to identify all eQTLs in low LD ($r^2 < 0.05$) with the best SNP, and again selected the best SNP. This procedure was performed iteratively to create a list of independent eQTLs for the gene/tissue pair. To reduce the number of spurious co-localizations, we only report variants in high LD with the FA lead variants that represented an independent eQTLs for the respective gene/tissue pair. Fine mapping and functional assessment of the cytokine gene cluster at 5q31.1 and the HLA region at 6p21 were performed in detail previously[14, 68], therefore, we only report proxy SNPs if they were predicted to have a "high" or "moderate" impact on the functionality of the resulting protein by the Ensembl Variant Effect Predictor[22]. The lead SNP at 1q21 (FLG) was excluded, since we demonstrate that the signal was due to LD with the known loss-of-function variants in FLG.

**Estimating the heritability explained by the identified loci**. The overall SNP-based heritability was estimated with LD score regression[69]. From the GWAS results on food allergy, we used a subset of 1.2 million HapMap SNPs. In order to quantify the heritability on the liability-scale the population prevalence was set to 5%. We then adjusted the GWAS results for the effects of the five lead variants (rs12123821 on chr. 1, rs11949166 on chr. 5, rs9273440 on chr. 6, rs2212434 on chr. 11, and rs12964116 on chr. 18) identified in our study. Again, we estimated the SNP-based heritability using the adjusted GWAS results. The heritability explained by the identified lead SNPs was calculated as the difference between the unadjusted heritability and the adjusted heritability.

**Data availability**. The data that support the findings of this study are included in Supplementary Data 1a–d. Additional information is available from the corresponding author upon reasonable request.

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

## Acknowledgements

We thank Christina Flachmeier, Theresa Thuß for excellent technical assistance, and Sylke Rietz and Marieke Hillen for sample collection and data management. SHIP is part of the Community Medicine Research net of the University of Greifswald, Germany, which is funded by the Federal Ministry of Education and Research (grant nos. 01ZZ9603, 01ZZ0103, and 01ZZ0403), the Ministry of Cultural Affairs as well as the Social Ministry of the Federal State of Mecklenburg-West Pomerania, and the network 'Greifswald Approach to Individualized Medicine (GANI_MED) funded by the Federal Ministry of Education and Research (grant 03IS2061A). Genome-wide data have been supported by the Federal Ministry of Education and Research (grant no. 03ZIK012).

## Author contributions

Y.-A.L. and K.Be. directed the study. B.K., K.Bl., R.S., N.Ha., M.P., G.H., J.S., H.R., S.Y., S.T., B.N., K.Be. and Y.-A.L. evaluated patients, collected samples, and contributed clinical data. G.H., C.O.S. and M.M.N. contributed control samples and genotypes. I.M., S.G., M.M.N., N.H. and Y.-A.L. contributed experimental data. I.M., S.G., F.R., X.H. and X.W. performed data analyses. I.M., S.G. and Y.-A.L. drafted the manuscript, all authors reviewed, revised, and approved the final manuscript.

## Additional information

**Competing interests:** The authors declare no competing financial interests.

