## [Transparent Peer Review · Nature Communications]

Reviewer #1 (Remarks to the Author):

In this manuscript by Marenholz and colleagues, the authors summarize results from the first GWAS focused specifically on food allergy diagnosed by oral food challenge (as opposed to sensitization, etc.) in a large German pediatric sample of food allergics and a mixture of children and adult controls. The team identifies a very compelling locus on chr 18q21.3, the SERPINB gene cluster, which is novel and replicated in an independent sample. They also identify significant associations in the FLG gene, the cytokine gene cluster on chr 5q, and the HLA locus. Of particular interest, the FLG association was independent of eczema status. Although the group leverages public databases to identify potential functional variants, this effort was limited to the top 2 SNPs in the SERPINB gene cluster, and overall the study falls short of demonstrating any other functional validation of their findings. Nevertheless, this manuscript advances our understanding of the genetics of food allergy.

Major Concerns:

1. The Bonferroni corrected P-value should be clearly stated in the Methods section; i.e., on pg 7 the authors refer to a significance level of 5.7×10^{-6} but with the HumanOmniExpress/number of tests the threshold should be lower.
2. Of interest, the association at the serpin locus was consistent for any food allergy. It is difficult to reconcile why, outside of the HLA locus, an association would be significant for a particular food allergen but not another. Is there a relationship with disease severity? For example, are peanut allergic children in this study more severe than children allergic to hen or milk? Could the authors please address this conundrum in the Discussion.
3. Why did the group not perform eQTL analyses for loci other than the SERPINB locus?
4. In the introduction the group references a prior study indicating a heritability of $\sim 80\%$ for peanut allergy. Did they consider heritability estimates in the current study?
5. It is not entirely clear in the Methods section whether or not the controls were unphenotyped, but it is to be assumed that they were. How might the unphenotyped status have impacted the findings?

Minor Concerns:

1. The Introduction is excessive and can be shortened.
2. Could the authors verify whether or not the same panel of foods was used for the German and Chicago food allergy cases?

Reviewer #2 (Remarks to the Author):

This is a well written manuscript describing the first GWAS of food allergy where the phenotype of most cases was stringently diagnosed by oral food challenge. After follow-up replication in two additional samples, four loci achieved genome-wide (GW) significance for association with some type of food allergy, including a locus in the SERPINB gene cluster on chromosome 18 that has not been previously implicated in any allergic disease. The analyses used are mostly appropriate, and the figures and tables are well drafted and do a good job conveying the most important results.

I have only one major concern. Although it is not explicitly stated anywhere in the manuscript, it appears from the genotyping arrays listed in the Methods section (lines 339-340) and from a statement in the Results that all individuals in both the discovery and replication sets were confirmed to be of European ancestry by PCA (lines 116-117), that the GOFA replication set was genotyped with genome-wide coverage. Nevertheless, for some reason the authors of this study decided to limit the markers included in the replication stage to only those achieving a p-value of 1×10^{-5} or better in the discovery GWAS. Even among this limited subset, they further restricted analysis to a single marker per 1 Mb in regions where multiple markers achieved this threshold of significance.

I see no reason for these restrictions. The GOFA replication set should be imputed and fully

analyzed just like the discovery GWAS, followed by a fixed-effects meta-GWAS of the discovery and replication association results. Unlike the analysis strategy pursued by the authors, this approach has the possibility of uncovering highly suggestive or even GW-significant association signals in the combined analysis that weren't pursued because the signal didn't achieve the threshold of significance in the relatively small discovery GWAS (497 cases & 2387 controls). Also, even for regions detected by the discovery set, the markers selected may not necessarily be the ones that would give maximal association in the combined discovery and replication analyses. Traditionally, replication sets were restricted to a relatively small number of markers because of the high cost of GWAS arrays. Decreasing costs have made it more feasible to genotype the entire sample set on such arrays (as was apparently done in this study), and in such cases all available data should be analyzed.

The other issues to be addressed are of more modest importance and are listed in the order of their appearance in the manuscript.

Abstract

(lines 43-45) This sentence mistakenly implies that a GWAS study was performed for over 900 affected children and 3,330 controls. Instead, as made clearer later in the manuscript, the GWAS (discovery) phase of the study involved only 497 cases and 2387 controls. However, if the authors re-analyze their data as a meta-GWAS as I suggest above, this sentence can stand as written.

(lines 50-51) As explained in more detail below, this sentence about variants in chromosome 18q21.3 influencing expression of several SERPINB genes implies causality. If the lead 18q21.3 SNPs are not in near-perfect or better LD with the best eQTL signals for these SERPINB genes, then his claim should be completely removed. If the lead SNPs can be demonstrated to be in near-perfect LD with the best eQTL variants, then this claim can remain but "influence" should be changed to "may influence" since further experimental work would be needed to determine which variant in the LD cluster is causal.

Results

(line 102) For the discovery sample 497 of the 522 ascertained cases passed QC, but every one of the 2387 ascertained controls did. Is there any reason for this difference (e.g., were some of the cases lower quality DNA than controls?).

(line 104) "Michigan University" should be "the University of Michigan"

(lines 112-113) When multiple SNPs from the same region were associated at $P < 1 \times 10^{-5}$ with disease in the discovery GWAS, 1/Mb was selected. It would have been better to use LD rather than physical spacing for selection, since independent signals in the same locus or even in entirely different loci could exist within a single 1 Mb region. However, the best approach would be to meta-analyze the entire imputed marker set for the GOFA replication sample with the discovery sample, as discussed above.

(lines 125-131) Association results for any FA for 1q21.3 and 5q31.1 lead SNPs are given for the GOFA discovery GWAS and GOFA replication set. Please also list meta-results (OR and p-value) for the combined GOFA discovery and replication associations.

(lines 157-158) It is noted that two independent signals in the cytokine gene cluster were detected. According to Supplementary Table 4, the p-value for rs2074369 changed by five orders of magnitude (10^{-10} to 10^{-5}) after conditioning on rs11949166 despite the low LD ($r^2=0.04$) between the two. This is an unexpected result. If LD was computed based on an independent reference set from the same population, it might be helpful to look at the squared correlation of the imputed dosages for these two SNPs in the discovery GWAS to see if it is significantly greater

than $r^2 = 0.04$, which might help explain this observation.

(lines 162-179) In this paragraph association results for two 18q21.3 association signals in low LD with each other (rs12964116 and rs1243064) are provided for the GOFA discovery GWAS and for the GOFA and Chicago Food Allergy replication sets. For rs12964116, GW-significance was achieved only for the meta-analysis of all three sample sets. For rs1243064, on the other hand, GW-significance was achieved for the GOFA discovery + replication meta-analysis. The association results for rs1243064 when meta-analyzing all three datasets are not given, presumably because they do not achieve GW-significance. However, in this situation it is not proper to pick and choose which meta-analysis results to present based on their p-value (i.e., 3-set meta-analysis results for rs12964116 vs. 2-set meta-analysis results for rs1243064). Because the 3-set meta-analysis has the most power to detect association signals, its results should be provided for both SNPs.

(lines 194-197) SNP rs1243064 is described as a tissue-specific eQTL whose risk allele decreases SERPINB10 expression in whole blood and increases SERPINB11 expression in esophageal mucosa. Although the eQTL results are interesting and worth noting, this sentence needs to be tempered. Many eQTL signals in current datasets such as GTEx are enormously significant, with some gene/tissue combinations having many highly significant eQTLs, including SNPs that are only in relatively modest LD with the best eQTL variant for a given gene/tissue combination. As detailed in my comments about Supplementary Table 6 below, it is important to compare the strength and significance of the rs1243064 eQTL signal with the best known eQTL variant for these two gene/tissue combinations. If rs1243064 is not in near-perfect LD with the strongest eQTL signal, then it is likely to be merely an LD shadow of the true causative eQTL variant, and this fact must be stated. Even if rs1243064 is in perfect or near-perfect LD with the strongest eQTL variant for these two tissue/gene combinations, without further experimental work one cannot be sure which of the LD-group is the actual causative variant, so words such as "decreasing" or "increasing" should be replaced with "negatively correlated with" and "positively correlated with".

Discussion

(lines 277-279) See pertinent comments concerning lines 194-197 in Results.

The total sample size of the GOFA discovery + replication sets of this study is quite small compared to recently published association studies for many other complex genetic traits. Accordingly, this study is only powered to detect loci of relatively large effect size, which is borne out by the fact that the ORs for the four GW-significant loci range from ~ 1.5 -4.0. Recruitment of larger food allergy patient samples for future association studies would almost surely discover additional loci with smaller effect sizes. Even though these loci will generally confer lower population attributable risk than the four loci detected by this study, they could still be helpful for understanding the biological mechanism of food allergies and for designing effective treatments. Perhaps a sentence or two about future directions can be included near the end of the Discussion to this effect.

Methods

(lines 339-340) Genotyping was performed on three different microarray platforms. Please add a description of which microarray was used for which dataset (GOFA discovery, GOFA replication, Chicago Food Allergy replication).

(lines 355-356) Clarify description of QC filters: "imputation quality $r^2 < 0.5$ " rather than " $r^2 < 0.5$ ", "HWE p-value $< 10^{-12}$ " rather than " $HWE < 10^{-12}$ ".

Table 3. Please add meta-analysis results (at a minimum p-values).

Figure 1. In the legend, the stated threshold for entering the replication phase (blue line) should

be $P < 1 \times 10^{-5}$ rather than $P < 1 \times 10^{-6}$.

Supplementary Information

Suppl. Fig 1: In figure legend, change "selection threshold" to "selection threshold for replication"

Suppl. Fig 3: In the legend, "(B)" seems misplaced; it probably should be placed immediately following the phrase "before (A) and after".

Suppl. Fig. 4: For panel B, it would be preferable to show a regional association plot where all variants are conditioned on the peak signal in panel A (rs11949166).

Suppl. Fig. 5: It would be helpful to add a panel B showing association after conditioning on rs1243064 to better demonstrate the independent association tagged by rs12964116.

Suppl. Tables 1A-1D: For all four of these tables, please add discovery + replication set meta-analysis results.

Suppl. Table 6: It would be very instructive to list p-values for the best eQTLs for SERPINB10 and SERPINB11 for all the various combinations of tissue and data source, and also the $LD-r^2$ between these best eQTLs and the SNPs currently listed in the table. This will allow evaluation if the study SNP could potentially be driving the observed eQTL effect or if it is only in imperfect LD with the causative eQTL variant.

Reviewer #3 (Remarks to the Author):

Marenholz et al report the results of a food allergy GWAS. They report the identification of 4 genome-wide significant loci, 3 novel for general food allergy. Strengths of the study include more strict definition of food allergy based on oral food challenge, good sample size compared to previous studies, use of a discovery and two independent replication cohorts, and stratification of results based on peanut, egg, and milk allergy. Overall, this work represents a good advancement of the field. The manuscript could be strengthened by addressing the following:

1. As noted by the authors, eczema is a significant co-morbidity of food allergy (approximately 80% of their cases) and findings from a food allergy GWAS could in part be driven by underlying eczema in food allergy cases. The authors attempt to control for this by stratifying the results for food allergy plus eczema compared to food allergy alone (Supplementary Table 3). This is a good start; however they do this only for the chromosome 1 and 5 SNPs—could they also show this for the HLA SNP and SERPINB7 SNPs? Secondly, the average age of ascertainment for their food allergy cases is around 2 years of age and the onset of eczema can occur beyond two years of age, often times until age 7-10 (and even adult-onset eczema). So again there is a question of whether the food allergy SNPs could be driven by eczema that may not have yet expressed itself. Do the authors have access to any eczema GWAS data in which further examination of the identified food allergy SNPs could be performed such that the relationship between the food allergy SNPs and eczema can be further clarified?
2. On page 7 the authors state in the text that a meta-analysis of all three studies resulted in a genome-wide significant p-value for rs12964116. Since this is an important claim could those results be fully shown (with allele frequencies, p-values, and ORs in each cohort and meta-analysis) in a supplementary table?
3. On page 7 the authors state that a second SERPINB SNP rs1243064 is associated with food allergy citing the low LD with rs12964116. Could the authors perform formal conditional analysis to show that the effect of rs1243064 is truly independent of rs12964116?
4. In the Discussion the authors suggest that filaggrin could act on food allergy risk through oral

mucosa rather than skin. Besides the fact that filaggrin is expressed in oral mucosa, can the authors cite any functional studies to support this claim?

Responses to the reviewers' comments:

Reviewer #1 (Remarks to the Author):

In this manuscript by Marenholz and colleagues, the authors summarize results from the first GWAS focused specifically on food allergy diagnosed by oral food challenge (as opposed to sensitization, etc.) in a large German pediatric sample of food allergics and a mixture of children and adult controls. The team identifies a very compelling locus on chr 18q21.3, the SERPINB gene cluster, which is novel and replicated in an independent sample. They also identify significant associations in the FLG gene, the cytokine gene cluster on chr 5q, and the HLA locus. Of particular interest, the FLG association was independent of eczema status. Although the group leverages public databases to identify potential functional variants, this effort was limited to the top 2 SNPs in the SERPINB gene cluster, and overall the study falls short of demonstrating any other functional validation of their findings. Nevertheless, this manuscript advances our understanding of the genetics of food allergy.

Major Concerns:

1. The Bonferroni corrected P-value should be clearly stated in the Methods section; i.e., on pg 7 the authors refer to a significance level of 5.7×10^{-6} but with the HumanOmniExpress/number of tests the threshold should be lower.

Indeed, the significance thresholds were not stated in the methods section. We have now added the *P*-value thresholds used in this study. In addition, the study design, including the *P*-value thresholds for each FA phenotype is illustrated in a new supplementary figure (Supplementary Fig. 1).

This paragraph now reads (p. 10, lines 188-194):

“Another new susceptibility locus which was previously not linked to any allergic disease was identified in chromosomal region 18q21.3 (Supplementary Fig. 8A). SNP rs12964116 located in intron 1 of *SERPINB7* (serpin peptidase inhibitor, clade B, member 7) was associated with food allergy in the GOFA discovery set (OR, 1.9; $P = 5.7 \times 10^{-6}$, Table 2) and replicated with the same risk allele and a similar effect size in the GOFA replication set (OR, 1.69; $P = 9.4 \times 10^{-3}$). Since rs12964116 did not reach the Bonferroni corrected *P*-value in the replication set ($P < 5.9 \times 10^{-5}$), we investigated the Chicago Food Allergy Study¹⁴ in order to confirm this locus.”

Methods (p. 21, lines 418-423): “In the GOFA replication set, significant association was defined as an association with the same risk allele as in the discovery set at the Bonferroni corrected *P*-value (0.05 / number of SNPs tested for a given phenotype). The Bonferroni corrected significance

thresholds were $P < 5.9 \times 10^{-5}$ for any FA, $P < 5.6 \times 10^{-5}$ for HE, $P < 6.1 \times 10^{-5}$ for PN, and $P < 5.9 \times 10^{-5}$ for CM. The threshold for genome-wide significance was $P < 5 \times 10^{-8}$ in the meta-analysis (Supplementary Fig. 1).”

2. Of interest, the association at the serpin locus was consistent for any food allergy. It is difficult to reconcile why, outside of the HLA locus, an association would be significant for a particular food allergen but not another. Is there a relationship with disease severity? For example, are peanut allergic children in this study more severe than children allergic to hen or milk? Could the authors please address this conundrum in the Discussion.

Reviewer 1 points out that one may not have expected an allergen-specific association outside the HLA locus due to its known role in binding and presenting exogenous proteins to initiate specific antibody responses as observed in allergy. He/she suspected that the finding of a peanut-locus may rather be related to food allergy severity because peanut allergy tends to be more severe than hen’s egg or cow’s milk allergy. Unfortunately, severity data are unavailable in our study.

We have therefore toned down the allergen-specific findings in the discussion:

Discussion (p. 18, lines 374-376): “In the context of limited study power, the finding of allergen-specific associations should be interpreted cautiously since the lack of association between an allergen-specific SNP and other FAs may be due to limited sample size of the subgroups.”

3. Why did the group not perform eQTL analyses for loci other than the SERPINB locus?

This is a valid point, we have now performed eQTL analyses for all variants reaching genome-wide significance. This information has been added to Supplementary Table 8.

4. In the introduction the group references a prior study indicating a heritability of ~80% for peanut allergy. Did they consider heritability estimates in the current study?

There are few studies estimating heritability of food allergy. The American study we quoted was based on 75 twin pairs and estimated the heritability of peanut allergy at 0.82 (95% CI 0.41–0.99). We have now added another study based on >25,000 Swedish twins aged 9-12 years, including >2,000 children

with food allergy. Their estimate of food allergy heritability of 0.78 (95% CI 0.74–0.82) was consistent with the previous report, but was founded on a much larger sample. We have now added this reference. We have now used LD score regression to obtain an estimate of food allergy heritability from our GWAS summary statistics. We now report the heritability estimate and the variance in food allergy liability explained by the 5 loci identified in this study. This information is provided in a new Supplementary Table 9.

Results (p. 12, lines 230-233): “We then used LD-score regression analysis in order to quantify the liability-scale heritability of food allergy that was explained by the lead variants identified in our study. Altogether, the food allergy susceptibility loci identified in this study explained approximately 10.2% of the variance in liability (Supplementary Table 9).”

Discussion (p. 17, lines 357-363): “We estimated the FA heritability from our GWAS summary statistics to be 24.4%. This is in contrast to previous estimates from twin studies which yielded estimates around 80%. Similar discrepancies have been reported for other complex diseases.⁵⁵ The difference between the heritability estimates from GWAS and from pedigree or twin studies may be due to an underestimation of the contribution of common environmental factors in twin studies, gene-environment interactions or model misspecification.⁵⁶ The five lead variants of the food allergy susceptibility loci identified in this study explained 10.2% of the heritability.”

Methods (p. 25, lines 511-519): “Estimating the heritability explained by the identified susceptibility loci. The overall SNP-based heritability was estimated with LD score regression.⁶⁹ From the GWAS results on food allergy, we used a subset of 1.2 million HapMap SNPs. In order to quantify the heritability on the liability-scale the population prevalence was set to 5%. We then adjusted the GWAS results for the effects of the 5 lead variants (rs12123821 on chr. 1, rs11949166 on chr. 5, rs9273440 on chr. 6, rs2212434 on chr. 11, and rs12964116 on chr. 18) identified in our study. Again, we estimated the SNP-based heritability using the adjusted GWAS results. The heritability explained by the identified lead SNPs was calculated as the difference between the unadjusted heritability and the adjusted heritability.”

5. It is not entirely clear in the Methods section whether or not the controls were unphenotyped, but it is to be assumed that they were. How might the unphenotyped status have impacted the findings?

The controls were population-based and NOT phenotyped for food allergy. We have now added this information in the description of the study populations of the methods section:

(p. 20, lines 399-404). “2,682 German control individuals without information on food allergy originated from the Heinz Nixdorf Recall Study (HNR). HNR is a population-based cohort study for cardiovascular disease¹⁶ comprising 4,800 individuals from Ruhr area in Germany. The GOFA replication set comprised another 380 children with food allergy and 986 unphenotyped control individuals from the Study of Health in Pomerania (SHIP), a population-based cohort from North-Eastern Germany.^{18”}

Moreover, we have added a new paragraph on the limitations of our study including the impact of unphenotyped controls to the discussion section.

(p. 18, lines 381-384). “In contrast to the well phenotyped cases included in this study, there was no reliable information on food allergy available for the controls. Therefore, the presence of affected individuals among controls may have decreased the power of our study. Given a food allergy prevalence of about 5% in Western Europe, the loss of power was probably minor.”

Minor Concerns:

1. The Introduction is excessive and can be shortened.

We have shortened the Introduction from 581 to 501 words.

2. Could the authors verify whether or not the same panel of foods was used for the German and Chicago food allergy cases?

We now state that information on the same food allergies were available for GOFA and the Chicago Food Allergy Study. The description of the study populations now reads:

For the GOFA populations (p. 20, lines 398-399). “Apart from any food allergy (FA), we investigated the three most common food allergies against hen’s egg (HE), peanut (PN), and cow’s milk (CM).”

For the Chicago Food Allergy Study (p. 20, lines 404-407). “The second replication set, the Chicago Food Allergy Study, has previously been described in detail.¹⁴ This study included 671 food allergic children of European ancestry of whom 316, 291, and 217 were allergic against PN, CM, and HE, respectively.”

Reviewer #2 (Remarks to the Author):

This is a well written manuscript describing the first GWAS of food allergy where the phenotype of most cases was stringently diagnosed by oral food challenge. After follow-up replication in two additional samples, four loci achieved genome-wide (GW) significance for association with some type of food allergy, including a locus in the SERPINB gene cluster on chromosome 18 that has not been previously implicated in any allergic disease. The analyses used are mostly appropriate, and the figures and tables are well drafted and do a good job conveying the most important results.

I have only one major concern. Although it is not explicitly stated anywhere in the manuscript, it appears from the genotyping arrays listed in the Methods section (lines 339-340) and from a statement in the Results that all individuals in both the discovery and replication sets were confirmed to be of European ancestry by PCA (lines 116-117), that the GOFA replication set was genotyped with genome-wide coverage. Nevertheless, for some reason the authors of this study decided to limit the markers included in the replication stage to only those achieving a p-value of 1×10^{-5} or better in the discovery GWAS. Even among this limited subset, they further restricted analysis to a single marker per 1 Mb in regions where multiple markers achieved this threshold of significance.

I see no reason for these restrictions. The GOFA replication set should be imputed and fully analyzed just like the discovery GWAS, followed by a fixed-effects meta-GWAS of the discovery and replication association results. Unlike the analysis strategy pursued by the authors, this approach has the possibility of uncovering highly suggestive or even GW-significant association signals in the combined analysis that weren't pursued because the signal didn't achieve the threshold of significance in the relatively small discovery GWAS (497 cases & 2387 controls). Also, even for regions detected by the discovery set, the markers selected may not necessarily be the ones that would give maximal association in the combined discovery and replication analyses.

Traditionally, replication sets were restricted to a relatively small number of markers because of the high cost of GWAS arrays. Decreasing costs have made it more feasible to genotype the entire sample set on such arrays (as was apparently done in this study), and in such cases all available data should be analyzed.

It is correct that the GOFA discovery set and the GOFA replication set both were genotyped using SNP arrays. This was not obvious from the methods section. We have added detailed array information for all sets included in our study in a new supplementary table (Supplementary Table 11). In addition, we now

explicitly state that genotyping, genotype imputation, and quality control were performed in the same way for the discovery and the replication set.

(p.22, lines 447-455): “Samples were genotyped on Illumina’s HumanOmniExpressExome-8 v1.2, HumanOmniExpress-12 v1.0, or HumanOmni1-Quad v1 (Supplementary Table 11). For the discovery and the replication set, the same QC criteria were applied. Individuals with a call rate < 0.97 or with high heterozygosity (> 0.35) were excluded. Individual SNPs were filtered according to the following criteria: i) low call rate (< 0.96 in cases or controls) ii) low minor allele frequency (MAF) < 0.005 in cases or controls), iii) genotypes out of Hardy-Weinberg equilibrium (HWE, $P < 0.00001$ in cases or $P < 0.0001$ in controls). SNPs with a call rate lower than 0.99 were excluded if having a MAF < 0.05 or if they were out of HWE ($P < 0.001$). Only SNPs fulfilling the above mentioned QC were used in subsequent steps.”

R2 suggests a different selection method for candidate SNPs at a given locus that should be based on LD rather than physical distance to uncover additional, independent association signals. We agree that an LD based candidate SNP selection will uncover independent association signals. We describe this in more detail in the following paragraphs.

In addition, R2 suggests a different analysis approach: Instead of our current study design consisting of GOFA discovery, GOFA replication, and Chicago Food Allergy Study replication, he/she suggests to meta analyze the 2 GOFA data sets and to replicate only in the Chicago data set.

One of the key requirements in GWAS is independent replication. As discussed in the introduction and acknowledged by all 3 reviewers as a strength of our study, the phenotype definition is a critical point when studying food allergy. Of the available data sets, only GOFA discovery and GOFA replication were phenotyped using double-blind placebo controlled food challenges. It is therefore necessary to use a study design in which candidate loci are replicated in an independent replication set with state-of-the-art phenotypes.

However, we do agree with Reviewer 2 that the stringent P -value of 10^{-5} used in the discovery cohort and the position-based marker selection may have hampered discovery at an early stage. We have therefore decreased the discovery threshold by 2 orders of magnitude. We now use a very modest P value of $<10^{-3}$ for marker selection. Along with the LD based SNP selection strategy, this increased the number of candidate SNPs in the discovery stage about 40-fold for each FA phenotype. We thus provide a comprehensive list of candidate SNPs that now includes very moderate associations (revised

Supplementary Tables 1A-D). The new study design is now described in the methods section and in a new Supplementary figure 1.

(p. 20, lines 409-426): “For each phenotype under study (FA, HE, PN, CM), all SNPs with moderate association in the discovery set ($P < 1 \times 10^{-3}$) were identified (FA 7699 SNPs, HE 8,959 SNPs, PN 6,794 SNPs, CM 6,955 SNPs). To define a locus, we grouped all consecutive SNPs with $P < 1 \times 10^{-3}$ and a distance < 1 Mb to the next SNP (FA 611 loci, HE 634 loci, PN 595 loci, CM 612 loci). At each locus, we selected the SNP with the lowest P -value as lead SNP. To identify additional, independent association signals within each locus, we identified all SNPs in low LD with the lead SNP ($r^2 < 0.2$) and again selected the best SNP. Thus the number of additional LD-selected SNPs was 236, 262, 223, and 233 SNPs, and the total number of candidate SNPs selected in the GOFA discovery set were 847, 896, 818, and 845 for the phenotypes FA, HE, PN, CM, respectively.

In the GOFA replication set, significant association was defined as an association with the same risk allele as in the discovery set at the Bonferroni corrected P -value ($0.05 / \text{number of SNPs tested for a given phenotype}$). The Bonferroni corrected significance thresholds were $P < 5.9 \times 10^{-5}$ for any FA, $P < 5.6 \times 10^{-5}$ for HE, $P < 6.1 \times 10^{-5}$ for PN, and $P < 5.9 \times 10^{-5}$ for CM. The threshold for genome-wide significance was $P < 5 \times 10^{-8}$ in the meta-analysis (Supplementary Fig. 1).

Variants replicating at nominal significance ($P < 0.05$) in the GOFA replication set and yielding a $P < 10^{-6}$ in the meta-analysis of GOFA discovery and GOFA replication, but not reaching the Bonferroni corrected P -value, were additionally confirmed in the Chicago Food Allergy Study.”

Indeed, this strategy revealed an additional FA locus reaching genome-wide significance on chromosome 11q13.5. For the 4 FA loci we had reported in the original version of the manuscript, this approach identified the same lead SNPs or SNPs in high LD.

We included the extended candidate SNPs in Supplementary Tables 1A to 1D. We performed all subsequent analyses for the newly identified locus and included the results in the discussion section.

The other issues to be addressed are of more modest importance and are listed in the order of their appearance in the manuscript.

Abstract

(lines 43-45) This sentence mistakenly implies that a GWAS study was performed for over 900 affected children and 3,330 controls. Instead, as made clearer later in the manuscript, the GWAS (discovery)

phase of the study involved only 497 cases and 2387 controls. However, if the authors re-analyze their data as a meta-GWAS as I suggest above, this sentence can stand as written.

We have rephrased the sentence in the abstract:

(p. 3, lines 48-50): “Here we report the first genome-wide association study on food allergy diagnosed by oral food challenge in 497 cases and 2387 controls. Another 1051 cases and 2510 controls were included in the replication.”

(lines 50-51) As explained in more detail below, this sentence about variants in chromosome 18q21.3 influencing expression of several SERPINB genes implies causality. If the lead 18q21.3 SNPs are not in near-perfect or better LD with the best eQTL signals for these SERPINB genes, then his claim should be completely removed. If the lead SNPs can be demonstrated to be in near-perfect LD with the best eQTL variants, then this claim can remain but “influence” should be changed to “may influence” since further experimental work would be needed to determine which variant in the LD cluster is causal.

Please see our reply to the detailed comment below.

Results

(line 102) For the discovery sample 497 of the 522 ascertained cases passed QC, but every one of the 2387 ascertained controls did. Is there any reason for this difference (e.g., were some of the cases lower quality DNA than controls?).

We had not reported the numbers before QC, we have now included the numbers before and after QC for the control sets in the genotyping and quality control section under Methods, and in the new Supplementary Figure 1:

(p. 23, lines 458-461): “After quality control, the GOFA discovery and GOFA replication sets included 497 (before QC 523) and 379 (before QC 380) cases as well as 2,387 (before QC 2,682) and 984 (before QC 986) 1,526 controls, respectively.”

(line 104) “Michigan University” should be “the University of Michigan”

(p. 6, lines 108): We have replaced “Michigan University” with “the University of Michigan”.

(lines 112-113) When multiple SNPs from the same region were associated at $P < 1 \times 10^{-5}$ with disease in the discovery GWAS, 1/Mb was selected. It would have been better to use LD rather than physical spacing for selection, since independent signals in the same locus or even in entirely different loci could exist within a single 1 Mb region. However, the best approach would be to meta-analyze the entire imputed marker set for the GOFA replication sample with the discovery sample, as discussed above.

As discussed above, we have addressed both points. Candidate SNP were now selected based on LD. Furthermore, we changed the SNP selection P value threshold from $P < 1 \times 10^{-5}$ to $P < 1 \times 10^{-3}$.

(lines 125-131) Association results for any FA for 1q21.3 and 5q31.1 lead SNPs are given for the GOFA discovery GWAS and GOFA replication set. Please also list meta-results (OR and P -value) for the combined GOFA discovery and replication associations.

We have added the results for the meta-analysis (Table 2 and results section).

(p. 7, lines 141-143): “Meta-analysis of the two sets yielded highly significant associations at 1q21.3 and 5q31.1 (rs12123821; OR, 2.65; $P = 2.6 \times 10^{-15}$ and rs11949166; OR, 0.63; $P = 4.3 \times 10^{-17}$).”

(lines 157-158) It is noted that two independent signals in the cytokine gene cluster were detected. According to Supplementary Table 4, the p-value for rs2074369 changed by five orders of magnitude (10^{-10} to 10^{-5}) after conditioning on rs11949166 despite the low LD ($r^2=0.04$) between the two. This is an unexpected result. If LD was computed based on an independent reference set from the same population, it might be helpful to look at the squared correlation of the imputed dosages for these two SNPs in the discovery GWAS to see if it is significantly greater than $r^2 = 0.04$, which might help explain this observation.

The reported LD $r^2 = 0.04$ was based on the Europeans in the 1000G project. We have now calculated LD for the controls of the GOFA discovery set yielding only slightly larger LD values (r^2 , 0.08 and D' , 0.36). There is some correlation between the 2 LD groups which is responsible for the decrease of the association signals in the one LD group after conditioning on the lead SNP of the other LD group. Nonetheless, as shown in Supplementary Table 5 and Supplementary Figures 5 and 6, the residual association remains strong pointing to more than one susceptibility locus.

(lines 162-179) In this paragraph association results for two 18q21.3 association signals in low LD with each other (rs12964116 and rs1243064) are provided for the GOFA discovery GWAS and for the GOFA and Chicago Food Allergy replication sets. For rs12964116, GW-significance was achieved only for the meta-analysis of all three sample sets. For rs1243064, on the other hand, GW-significance was achieved for the GOFA discovery + replication meta-analysis. The association results for rs1243064 when meta-analyzing all three datasets are not given, presumably because they do not achieve GW-significance. However, in this situation it is not proper to pick and choose which meta-analysis results to present based on their p-value (i.e., 3-set meta-analysis results for rs12964116 vs. 2-set meta-analysis results for rs1243064). Because the 3-set meta-analysis has the most power to detect association signals, its results should be provided for both SNPs.

Table 2 contains the results for GOFA discovery, GOFA replication and the meta-analysis of the two sets. This table illustrates, which loci achieved genome-wide significance after the discovery stage and after the GOFA replication stage. We have added the ORs to the meta-analysis column. The SERPIN variant rs1243064 reached genome-wide significance in the meta-analysis for HE allergy, but only nominal significance in GOFA replication. It was therefore taken forward to the Chicago food allergy study where it did not replicate. This is clearly stated now:

(p. 10, lines 197-209): “Within the *SERPINB* gene cluster, a second SNP (rs1243064) in moderate LD with rs12964116 ($r^2 = 0.06$, $D' = 0.71$) was associated with food allergy (Supplementary Fig. 9A). In order to explore, whether the two SNPs represented independent association signals we mutually conditioned on the two lead variants (Supplementary Table 6). In both cases, association of the other variant with food allergy decreased but was still present suggesting more than one risk haplotype at this locus (Supplementary Fig. 8B and 9B, Supplementary Table 6).

Association of rs1243064 was confirmed in the GOFA replication set for hen’s egg allergy at nominal significance, reaching genome-wide significance in the meta-analysis of GOFA discovery and replication set ($P = 4.2 \times 10^{-8}$, Table 2). In the Chicago Food Allergy Study the same risk allele was identified (Table 3, Supplementary Table 7). However, association did not reach significance ($P = 0.15$) which may be due to reduced power in a small sample with a less stringent phenotype definition that was not based on OFCs.”

Table 3 contains the results for GOFA discovery, GOFA replication and the Chicago Food Allergy study. We have added another column now presenting the meta-analysis P-values for all 3 sets.

(lines 194-197) SNP rs1243064 is described as a tissue-specific eQTL whose risk allele decreases SERPINB10 expression in whole blood and increases SERPINB11 expression in esophageal mucosa. Although the eQTL results are interesting and worth noting, this sentence needs to be tempered. Many eQTL signals in current datasets such as GTEx are enormously significant, with some gene/tissue combinations having many highly significant eQTLs, including SNPs that are only in relatively modest LD with the best eQTL variant for a given gene/tissue combination. As detailed in my comments about Supplementary Table 6 below, it is important to compare the strength and significance of the rs1243064 eQTL signal with the best known eQTL variant for these two gene/tissue combinations. If rs1243064 is not in near-perfect LD with the strongest eQTL signal, then it is likely to be merely an LD shadow of the true causative eQTL variant, and this fact must be stated. Even if rs1243064 is in perfect or near-perfect LD with the strongest eQTL variant for these two tissue/gene combinations, without further experimental work one cannot be sure which of the LD-group is the actual causative variant, so words such as “decreasing” or “increasing” should be replaced with “negatively correlated with” and “positively correlated with”.

Reviewer 2 points out that association of genetic variants with clinical phenotypes and gene expression levels may occur coincidentally, and may not necessarily represent disease-relevant changes. The association of a disease-SNP with mRNA levels of a gene are probably not disease-relevant if other SNPs with much stronger effects on that gene’s expression level shows weaker association with the disease.

To address this problem, we have now used a more conservative approach. For the five FA-associated loci, we searched for eQTLs using the SNIPA tool (single nucleotide polymorphism annotator). If a SNP was associated with the expression of a gene in a tissue, we identified all independent eQTLs for that gene/tissue pair. We then created a list of independent eQTLs for that gene/tissue pair (as suggested by Zhernakova et al. Nat Genet 2017, PMID 27918533) and only report an eQTL if a FA associated lead SNP or its proxy ($LD\ r^2 > 0.8$) represented an independent eQTL for that gene/tissue pair.

With this more conservative approach, only one eQTL for rs1243064 influencing the Expression of SERPINB10 in blood was identified and reported. The others were removed. The method for eQTL selection is now described in the Methods section, and the results in Supplementary Tables 8) were changed accordingly.

(p. 24, lines 495-503): “To detect associations of the food allergy loci with gene expression levels, we used the single nucleotide polymorphism annotator (SNIPA)⁶⁷ tool to query publically available databases on expression quantitative trait loci (eQTL) in relevant tissues. If a SNP was associated with the expression of a gene in a tissue, we identified all independent eQTLs for that gene/tissue pair. To this end, we selected the best eQTL of the gene/tissue pair, then used LD link²¹ to identify all eQTLs in low LD ($r^2 < 0.05$) with the best SNP, and again selected the best SNP. This procedure was performed iteratively to create a list of independent eQTLs for the gene/tissue pair. To reduce the number of spurious co-localizations, we only report variants in high LD with the FA lead variants that represented an independent eQTLs for the respective gene/tissue pair.”

As suggested by Reviewer 2, we changed the wording to “negatively correlated with”.

(p. 11, lines 226-229): “The other lead SNP, rs1243064, was a tissue-specific expression quantitative trait locus (eQTL) with the risk allele rs1243064A negatively correlated with *SERPINB10* expression in whole blood (Supplementary Table 8).”

Discussion

(lines 277-279) See pertinent comments concerning lines 194-197 in Results.

The total sample size of the GOFA discovery + replication sets of this study is quite small compared to recently published association studies for many other complex genetic traits. Accordingly, this study is only powered to detect loci of relatively large effect size, which is borne out by the fact that the ORs for the four GW-significant loci range from ~1.5-4.0. Recruitment of larger food allergy patient samples for future association studies would almost surely discover additional loci with smaller effect sizes. Even though these loci will generally confer lower population attributable risk than the four loci detected by this study, they could still be helpful for understanding the biological mechanism of food allergies and for designing effective treatments. Perhaps a sentence or two about future directions can be included near the end of the Discussion to this effect.

We have added a paragraph on the limitations of our study including the power issue and on future directions in genetic studies of food allergy to the discussion section.

(p. 18, lines 364-373): “Some limitations of our study need to be discussed. Although we performed the largest GWAS on food allergy to date, the sample size was relatively small for a GWAS on a complex genetic trait. This was due to the strict phenotype definition used in our study. Though recommended in current guidelines,^{3, 15} oral food challenges are not always performed as the diagnostic standard method. As a consequence for challenge-proven food allergy, large numbers of patients are difficult to obtain. This study was powered to detect loci of relatively large effect sizes. While the combined GOFA set had a power of 99% to detect a risk variant with an OR of 1.6 (using an allelic model with 5% prevalence, 20% risk allele frequency, and $\alpha < 5 \times 10^{-8}$),⁵⁷ the power dropped to 10% for variants with a moderate OR of 1.3. Additional studies in larger samples will be required to evaluate variants with low effect sizes.”

Methods

(lines 339-340) Genotyping was performed on three different microarray platforms. Please add a description of which microarray was used for which dataset (GOFA discovery, GOFA replication, Chicago Food Allergy replication).

We have added a supplementary table (Supplementary Table 11) describing the arrays used for each of the data sets in more detail.

(lines 355-356) Clarify description of QC filters: “imputation quality $r^2 < 0.5$ ” rather than “ $r^2 < 0.5$ ”, “HWE p-value $< 10^{-12}$ ” rather than “HWE $< 10^{-12}$ ”.

We have added “imputation quality” and “P-value”.

Table 3. Please add meta-analysis results (at a minimum p-values).

We have added the meta-analysis P-values to Table 3. In addition, we have added meta-analysis P-values for all 3 study populations for the new candidate loci which were selected due to the modified study design (Supplementary Table 2).

Figure 1. In the legend, the stated threshold for entering the replication phase (blue line) should be $P < 1 \times 10^{-5}$ rather than $P < 1 \times 10^{-6}$.

In the figure legend, we have replaced $P < 1 \times 10^{-6}$ with the correct P-value threshold of $P < 1 \times 10^{-3}$.

Supplementary Information

Suppl. Fig 1: In figure legend, change “selection threshold” to “selection threshold for replication”

In the legend of Supplementary Fig 1, we have changed “selection threshold” to “selection threshold for replication”

Suppl. Fig 3: In the legend, “(B)” seems misplaced; it probably should be placed immediately following the phrase “before (A) and after”.

Thank you, this has been corrected.

Suppl. Fig. 4: For panel B, it would be preferable to show a regional association plot where all variants are conditioned on the peak signal in panel A (rs11949166).

We have included the conditional analysis for the susceptibility locus on chromosome 5q31.1 (Supplementary Figures 5 and 6). We performed the analysis for the 2 lead SNPs, rs11949166 and rs2074369, which are in low LD. For both LD groups, panel B clearly shows a significant effect after conditioning for the lead SNP of the other group.

Suppl. Fig. 5: It would be helpful to add a panel B showing association after conditioning on rs1243064 to better demonstrate the independent association tagged by rs12964116.

We have added the conditional analysis for the susceptibility locus on chromosome 18q21.3 (Supplementary Table 6, Supplementary Figures 8 and 9). Again, we performed the analysis for the 2 lead SNPs, rs12964116 and rs1243064 which are in moderate LD ($r^2 = 0.06$, $D' = 0.71$). After conditioning for rs1243064, association of rs12964116 is weaker but still present. Due to moderate r^2 between the 2 lead SNPs both association signals are slightly correlated. In order not to miss any potentially relevant SNP in this locus, we report both SNPs. We have replaced “low LD” with “moderate LD” (page 10, line 198, page 16, line 322).

Suppl. Tables 1A-1D: For all four of these tables, please add discovery + replication set meta-analysis results.

We have added the meta-analysis results for all candidate variants in Supplementary Tables 1A to 1D. As described above, the number of candidate variants has been increased about 40-fold, due to the less conservative *P*-value used for marker selection in the GOFA discovery set.

Suppl. Table 6: It would be very instructive to list *p*-values for the best eQTLs for SERPINB10 and SERPINB11 for all the various combinations of tissue and data source, and also the LD- r^2 between these best eQTLs and the SNPs currently listed in the table. This will allow evaluation if the study SNP could potentially be driving the observed eQTL effect or if it is only in imperfect LD with the causative eQTL variant.

Please see our detailed comments above. In Suppl. Table 8, we now report only those food allergy associated SNPs which are in high LD with the best SNP of an independent eQTL. Only 1 eQTL remained, all others were removed.

Reviewer #3 (Remarks to the Author):

Marenholz et al report the results of a food allergy GWAS. They report the identification of 4 genome-wide significant loci, 3 novel for general food allergy. Strengths of the study include more strict definition of food allergy based on oral food challenge, good sample size compared to previous studies, use of a discovery and two independent replication cohorts, and stratification of results based on peanut, egg, and milk allergy. Overall, this work represents a good advancement of the field. The manuscript could be strengthened by addressing the following:

1. As noted by the authors, eczema is a significant co-morbidity of food allergy (approximately 80% of their cases) and findings from a food allergy GWAS could in part be driven by underlying eczema in food allergy cases. The authors attempt to control for this by stratifying the results for food allergy plus eczema compared to food allergy alone (Supplementary Table 3). This is a good start; however they do this only for the chromosome 1 and 5 SNPs—could they also show this for the HLA SNP and SERPINB7 SNPs?

We had performed the eczema-stratified analysis only for known AD loci. We have now added the eczema-stratified results for all identified susceptibility loci (Supplementary Table 4).

Secondly, the average age of ascertainment for their food allergy cases is around 2 years of age and the onset of eczema can occur beyond two years of age, often times until age 7-10 (and even adult-onset eczema). So again there is a question of whether the food allergy SNPs could be driven by eczema that may not have yet expressed itself.

Reviewer 3 raised the concern that the prevalence of eczema may have been underestimated in our study sample due to the young age at recruitment. We assessed this in our GOFA study sets. The mean age at food allergy diagnosis was 2.1 and 2.8 years in the GOFA discovery and GOFA replication sets. However, follow-up data was available in a large number of cases, the mean age at last follow-up was 69 months. Of 717 GOFA cases with eczema, 590 (82.3 %) had age of eczema onset available, including 381 children aged 0-5 years, 140 children aged 6-10 years and 69 children >10 years. The mean age of eczema onset was 4.5 months (95%CI 3.9-4.6 months) among 0-5 year olds, 4.6 months (95%CI 4.0-5.3 months) among 5-10 year olds, and 7.2 months (95%CI 4.1-10.2 months) among >10 year olds. The maximum age of onset observed in children >10 years was 48 months.

Although eczema may develop in later childhood or adulthood, this seems to be a rare event in children with food allergy. Of 69 children with follow-up >10 years of age (mean 13.1 years, 95%CI 12.6-13.7 years), 92.7% manifested eczema within the first year of life, and 95.7% within the first 2 years of life. New onset eczema was not observed after the age of 48 months. The mean age at last follow-up in children without eczema was 81 months. It is therefore unlikely that the proportion of eczema among food allergic children was underestimated. This information has now been added to the Supplementary Information. We refer to it in the phenotype description of eczema:

(p. 22, lines 441-444): “A physician’s diagnosis of eczema was made according to standard criteria in the presence of a chronic or chronically relapsing pruritic dermatitis with the typical morphology and distribution.^{58, 59} More detailed information on eczema onset in the GOFA study sets is provided in the Supplementary Note 1.”

Do the authors have access to any eczema GWAS data in which further examination of the identified food allergy SNPs could be performed such that the relationship between the food allergy SNPs and eczema can be further clarified?

While four of the five FA susceptibility loci reported here, were previously identified in GWAS on eczema, the SERPINB locus was not found to be associated with eczema or any other allergic disease (Welter D, *et al.*, *Nucleic Acids Res* **42**, D1001-D1006 (2014). PMID 24316577. In our eczema stratified analysis, we demonstrated an effect on food allergy independent of eczema for four of five loci. The other locus on chromosome 11q13.5 revealed a residual effect on FA in children without eczema. Due to the relatively small number of food allergic children without eczema (n = 152), this effect was no longer significant. It is, however, noteworthy that the effect size for food allergy was much larger (OR 1.35) than the effect size reported in the largest eczema GWAS (OR 1.09). Thus, larger studies are required to demonstrate an effect of these loci on food allergy independent of eczema. We have included this point in the discussion.

(p. 15, lines 313-320): “The *C11orf30/LRRC32* region is a known risk locus for eczema,³⁶ asthma,^{37, 38} and other inflammatory diseases such as inflammatory bowel disease.³⁹ The lead SNP rs2212434 has previously been identified in the largest meta-GWAS on eczema, in which the odds ratio was 1.09 (95% CI, 1.07–1.12).²⁰ For food allergy, we estimated an OR of 1.35 (95% CI, 1.20-1.51) which was even higher when considering the combined food allergy plus eczema phenotype (OR, 1.40; 95% CI, 1.25-1.58). Since *C11orf30/LRRC32* was also strongly associated with the atopic march (rs2155219, OR, 1.33; 95% CI, 1.24–1.43),⁴⁰ our results support a key role of this locus in the development of multiple allergic disorders including food allergy.”

2. On page 7 the authors state in the text that a meta-analysis of all three studies resulted in a genome-wide significant p-value for rs12964116. Since this is an important claim could those results be fully shown (with allele frequencies, p-values, and ORs in each cohort and meta-analysis) in a supplementary table?

We have added the *P*-values for the meta-analysis of all 3 cohorts to Table 3. This table also contains the allele frequencies and *P*-values for the different study populations. Since ORs are not available for the Chicago Food Allergy Study, we report the ORs for the GOFA discovery and replication sets in Table 2.

3. On page 7 the authors state that a second SERPINB SNP rs1243064 is associated with food allergy citing the low LD with rs12964116. Could the authors perform formal conditional analysis to show that the effect of rs1243064 is truly independent of rs12964116?

We have added the conditional analysis for the susceptibility locus on chromosome 18q21.3 (Supplementary Table 6, Supplementary Figures 8 and 9). Again, we performed the analysis for the 2

lead SNPs, rs12964116 and rs1243064 which are in moderate LD ($r^2 = 0.06$, $D' = 0.71$). After conditioning for rs1243064, association of rs12964116 is weaker but still present. Due to moderate r^2 between the 2 lead SNPs both association signals are slightly correlated. In order not to miss any potentially relevant SNP in this locus, we report both SNPs. We have replaced “low LD” with “moderate LD” (page 10, line 198, page 16, line 322).

4. In the Discussion the authors suggest that filaggrin could act on food allergy risk through oral mucosa rather than skin. Besides the fact that filaggrin is expressed in oral mucosa, can the authors cite any functional studies to support this claim?

We have extended the discussion on a potential barrier defect in the oral mucosa due to filaggrin deficiency. Accordingly, we have added references linking filaggrin deficiency to an impaired epithelial barrier in the esophagus.

(p. 14, lines 292-298): “Interestingly, filaggrin is expressed in the oral and esophageal mucosa, but not in airway epithelia.^{29, 30} While in eczema-associated asthma allergic sensitization through the defective skin barrier seems to play a pivotal role in disease development, in food allergy, enhanced penetration of allergens may occur through a leaky epithelial barrier in the upper gastrointestinal tract independently of the skin. A recent study suggested a link between down-regulation of filaggrin in the esophageal mucosa and impairment of the corresponding epithelial barrier.³⁰”

Reviewer #1 (Remarks to the Author):

The authors have addressed a prior concern of lack of specificity of the threshold of significance, and the manuscript is strengthened by performing eQTL analyses for all variants reaching genome-wide significance instead of only for the SERPINB locus. This team's new observation of a potential underestimation of the contribution of non-genetic factors in risk of FA vis a vis the heritability estimates that have been added to the manuscript, and estimates that the 5 top SNPs explain 10.2% of the heritability, is an added strength overall. Findings of an allergen-specific association outside the HLA locus in the absence of functional validation studies remain a concern, although the authors have tempered their interpretation of this observation in the Discussion section. The authors are to be commended for their responsiveness to the other reviewers' comments and overall this manuscript reads sufficiently stronger than the original version. Plans for future studies and a process for following up on discoveries presented here remain vague, but overall this manuscript is substantially improved.

Reviewer #2 (Remarks to the Author):

The authors are to be commended for their detailed responses to all of the many suggestions made by the reviewers and for the several newly run analyses and the extensive editing of the original manuscript that this entailed. In general, I believe that all points raised in the first round of review have been satisfactorily addressed.

As regards my major concern that the original association analysis used only a small subset of the markers typed for the replication set, I am mostly satisfied by the authors' remedy of lowering their GWAS p-value threshold from 1×10^{-5} to 1×10^{-3} for entry into the replication analysis, as well as using an LD-based rather than position-based approach to select independent variants. It is gratifying that this alternative analysis yielded an additional FA locus. However, I do still believe that a joint analysis of all imputed markers passing the $r^2 > 0.5$ imputation quality threshold in both the GOFA discovery and replication sets would have been preferable. Several pioneering methodological studies (Skol et al., 2006, Nat Genet 38:209-213; Kraft et al., 2009, Stat Sci 24:561-573; Thomas et al., 2009, Stat Sci 24:414-429) have outlined the reasons why a two-stage approach (discovery GWAS followed by independent replication of a small subset of the most promising discovery signals) is really just a cost-effective alternative to a single-stage discovery analysis that uses the entire available sample for GWAS, and that robust replication actually requires independent replication by a different investigative team in a different population using different technologies and designs. Nevertheless, it is unlikely that inclusion of the full replication set would have materially changed the results presented in the revised manuscript.

Reviewer #3 (Remarks to the Author):

The authors have improved the manuscript with their revisions and addressed this reviewer's concerns.

REVIEWERS' COMMENTS:

Reviewer #1 (Remarks to the Author):

The authors have addressed a prior concern of lack of specificity of the threshold of significance, and the manuscript is strengthened by performing eQTL analyses for all variants reaching genome-wide significance instead of only for the SERPINB locus. This team's new observation of a potential underestimation of the contribution of non-genetic factors in risk of FA vis a vis the heritability estimates that have been added to the manuscript, and estimates that the 5 top SNPs explain 10.2% of the heritability, is an added strength overall. Findings of an allergen-specific association outside the HLA locus in the absence of functional validation studies remain a concern, although the authors have tempered their interpretation of this observation in the Discussion section. The authors are to be commended for their responsiveness to the other reviewers' comments and overall this manuscript reads sufficiently stronger than the original version. Plans for future studies and a process for following up on discoveries presented here remain vague, but overall this manuscript is substantially improved.

Reviewer 1 underlines that we have adequately addressed his/her concerns.

For future genetic studies, recruitment of larger populations using state of the art diagnostic methods is urgently required. Both points, the importance of a strict phenotype definition and the power limitations due to the sample size had been discussed in detail in the revised manuscript where we conclude that "additional studies in larger samples will be required to evaluate variants with low effect sizes" (p. 18) and "a strict phenotype definition, as set forth by recent medical guidelines on food allergy, is important for studies on the genetics of food allergy" (p.19). We have now added a statement on future functional studies for the new allergy locus (p.17, "Functional studies will be required to gain a better understanding of the physiological role of clade B serpins in food allergy.").

We thank the reviewer for his/her very helpful comments.

Reviewer #2 (Remarks to the Author):

The authors are to be commended for their detailed responses to all of the many suggestions made by the reviewers and for the several newly run analyses and the extensive editing of the original manuscript that this entailed. In general, I believe that all points raised in the first round of review have been satisfactorily addressed.

As regards my major concern that the original association analysis used only a small subset of the markers typed for the replication set, I am mostly satisfied by the authors' remedy of lowering their GWAS p-value threshold from 1×10^{-5} to 1×10^{-3} for entry into the replication analysis, as well as using an LD-based rather than position-based approach to select independent variants. It is gratifying that this alternative analysis yielded an additional FA locus. However, I do still believe that a joint analysis of all imputed markers passing the $r^2 > 0.5$ imputation quality threshold in both the GOFA discovery and replication sets would have been preferable. Several pioneering methodological studies (Skol et al., 2006, Nat Genet 38:209-213; Kraft et al., 2009, Stat Sci 24:561-573; Thomas et al., 2009, Stat Sci 24:414-429) have outlined the reasons why a two-stage approach (discovery GWAS followed by independent replication of a small subset of the most promising discovery signals) is really just a cost-effective alternative to a single-stage discovery analysis that uses the entire available sample for GWAS, and that robust replication actually requires independent replication by a different investigative team in a different population using different technologies and designs. Nevertheless, it is unlikely that inclusion of the full replication set would have materially changed the results presented in the revised manuscript.

We thank the reviewer for his/her very helpful comments. He/she is now satisfied with our analytical approach. His/her comments were instrumental in uncovering the additional chromosome 11 locus for food allergy locus.

Reviewer #3 (Remarks to the Author):

The authors have improved the manuscript with their revisions and addressed this reviewer's concerns.

We thank all three reviewers for their comments and their positive assessment of our work.